# Dendritic spikes in hippocampal granule cells are necessary for long-term potentiation at the perforant path synapse

Sooyun Kim[1,2]*, Yoonsub Kim[1], Suk-Ho Lee[1,2], Won-Kyung Ho[1,2]*

[1]Department of Physiology, Seoul National University College of Medicine, Seoul, Korea; [2]Neuroscience Research Institute, Seoul National University College of Medicine, Seoul, Korea

**Abstract** Long-term potentiation (LTP) of synaptic responses is essential for hippocampal memory function. Perforant-path (PP) synapses on hippocampal granule cells (GCs) contribute to the formation of associative memories, which are considered the cellular correlates of memory engrams. However, the mechanisms of LTP at these synapses are not well understood. Due to sparse firing activity and the voltage attenuation in their dendrites, it remains unclear how associative LTP at distal synapses occurs. Here, we show that NMDA receptor-dependent LTP can be induced at PP-GC synapses without backpropagating action potentials (bAPs) in acute rat brain slices. Dendritic recordings reveal substantial attenuation of bAPs as well as local dendritic $Na^+$ spike generation during PP-GC input. Inhibition of dendritic $Na^+$ spikes impairs LTP induction at PP-GC synapse. These data suggest that dendritic spikes may constitute a key cellular mechanism for memory formation in the dentate gyrus.
DOI: https://doi.org/10.7554/eLife.35269.001

*For correspondence:
sooyun.kim@snu.ac.kr (SK);
wonkyung@snu.ac.kr (W-KH)

**Competing interests:** The authors declare that no competing interests exist.

## Introduction

The cortico-hippocampal circuit is implicated in the formation, storage, and retrieval of spatial and episodic memories (*Lisman, 1999*). The dentate gyrus (DG), the first stage in the hippocampal circuitry, receives abundant excitatory projections from the entorhinal cortex via the perforant-path (PP) synapses. Theoretical models of hippocampal function propose that the DG is critically involved in pattern separation and that synaptic transmission and plasticity at PP-granule cell (GC) synapses in the DG is required to remove redundant memory representations (*Marr, 1971*; *McNaughton and Morris, 1987*; *Treves and Rolls, 1994*). In agreement with theoretical predictions, knockout of *N*-methyl-D-aspartate (NMDA) receptors in GCs impairs long-term potentiation (LTP) and the ability to rapidly form a contextual representation and discriminate it from previous similar memories in a contextual fear conditioning task (*McHugh et al., 2007*). Moreover, GCs that were activated by contextual fear conditioning, referred to as memory engram cells, present clear signatures of synaptic potentiation such as a larger AMPA-NMDA ratio and a greater density of dendritic spines (*Ryan et al., 2015*). Thus, knowledge of plasticity at PP-GC synapses is essential for understanding the hippocampal function.

Attempts to understand synaptic plasticity rules at PP-GC synapses date to studies of *Bliss and Lomo (1973)*, who first demonstrated LTP in the hippocampus by high-frequency stimulation of the PP fibers in vivo. More recently, theta-burst high-frequency stimulation (TBS) of the PP has been used to induce LTP at PP-GC synapses (*Schmidt-Hieber et al., 2004*; *McHugh et al., 2007*; *Ge et al., 2007*). However, the underlying mechanisms for the induction of LTP with TBS at these

synapses remain unclear. Associative forms of synaptic plasticity depend on a presynaptic activity (e.g. excitatory postsynaptic potential, EPSP) and a postsynaptic signal (e.g. action potential, AP). Classically, a backpropagating AP (bAP) provides the associative postsynaptic signal at the synaptic site for the induction of LTP (*Hebb, 1949*; *Magee and Johnston, 1997*; *Dan and Poo, 2006*; *Feldman, 2012*). However, axosomatic APs are poorly propagated back into the dendrites of the GCs (*Krueppel et al., 2011*) and are unlikely to contribute to the induction of LTP. In addition, mature GCs are relatively silent during exploration (*Schmidt-Hieber et al., 2014*; *Diamantaki et al., 2016*) and fire with a low number of APs. Thus, spike-timing-dependent plasticity (STDP) that depends on an axosomatic postsynaptic AP will rarely occur under natural conditions (*Feldman, 2012*). The pronounced attenuation of AP backpropagation, the low occurrence of APs, along with the distinct intrinsic features of mature GCs, such as a hyperpolarized resting membrane potential and reduced excitability (*Scharfman and Schwartzkroin, 1990*; *Mongiat et al., 2009*; *Pernía-Andrade and Jonas, 2014*), cannot explain how distal synaptic PP inputs can be potentiated. Resolving this question has critical implications for understanding both the mechanism and function of memory encoding and retrieval (*Ryan et al., 2015*).

An alternative postsynaptic signal that may contribute to LTP induction at PP synapses innervating distal dendrites of GCs is a locally generated dendritic spike (*Sjöström et al., 2008*). Synaptic plasticity at distal synapses may occur in the absence of axosomatic APs via dendritic sodium spikes, which provide an alternative source of postsynaptic depolarization necessary for LTP induction (*Golding et al., 2002*; *Kim et al., 2015*). In GCs, linear somatic responses can be seen when glutamate is applied in the dendrites (*Krueppel et al., 2011*). However, without directly accessing the electrical properties of the distal GC dendrites, it is still not known whether synaptic stimulation can generate local dendritic spikes in GCs.

To address this question, we performed subcellular patch-clamp recordings on the thin dendrites of GCs. We found that inhibition of dendritic $Na^+$ channels prevented LTP induction via TBS at PP-GC synapses. Because dendritic $Na^+$ channels could generate local spikes that were independent of axosomatic AP generation and were not affected by voltage attenuation, we suggest that $Na^+$ spikes in the dendrites provide the postsynaptic signal necessary for the induction of LTP at PP-GC synapses.

## Results

We examined GCs with an input resistance ($R_{in}$) <200 MΩ ($R_{in}$ = 108.4 ± 2.6 MΩ; resting membrane potential: –81.3 ± 0.2 mV; see 'Materials and methods') which corresponds to the mature GC population in the acute hippocampal slices from rats (*Schmidt-Hieber et al., 2004*).

### TBS-induced LTP at the PP-GC synapses does not require postsynaptic bAPs

We first induced long-term potentiation (LTP) in GCs by theta-burst stimulation (TBS) of the PP synapses in the outer third of the molecular layer (*Figure 1A*). To activate the PP synapses, the tip of a stimulation electrode was placed in close proximity to the dendrite (<50 μm) using fluorescence microscopy ('Materials and methods'). When bursts of postsynaptic APs were evoked during TBS, TBS-induced LTP caused an increase in the amplitude of the excitatory postsynaptic potentials (EPSPs; from 6.72 ± 0.71 mV to 12.69 ± 1.44 mV, n = 6, p<0.05; *Figure 1B*). However, axosomatic APs in GCs are substantially attenuated with distance from the soma (*Krueppel et al., 2011*), and therefore are not likely to provide the necessary depolarization at distal dendrites. To test whether axosomatic APs are required for the induction of LTP at these distal synapses, we either locally applied tetrodotoxin (TTX) to the proximal axon, soma, and proximal dendrites in a subset of experiments (6 of 13 experiments) or adjusted the stimulus intensity to prevent axosomatic AP initiation and backpropagation occur during TBS. The absence of axosomatic spikes did not block LTP induction, indicating that bAPs are not critical for this form of LTP (from 6.96 ± 0.40 mV to 9.96 ± 0.86 mV, n = 13, p<0.01; *Figure 1C*). Intriguingly, activation of the PP synapses during TBS often produced voltage responses with a fast depolarizing phase similar to somatic events observed during dendritic spike generation in other types of neurons (*Figure 1C,D*; *Golding and Spruston, 1998*; *Golding et al., 2002*; *Jarsky et al., 2005*; *Losonczy and Magee, 2006*; *Kim et al., 2012*). Consistent with previous results (*Golding and Spruston, 1998*; *Golding et al., 2002*; *Losonczy and*

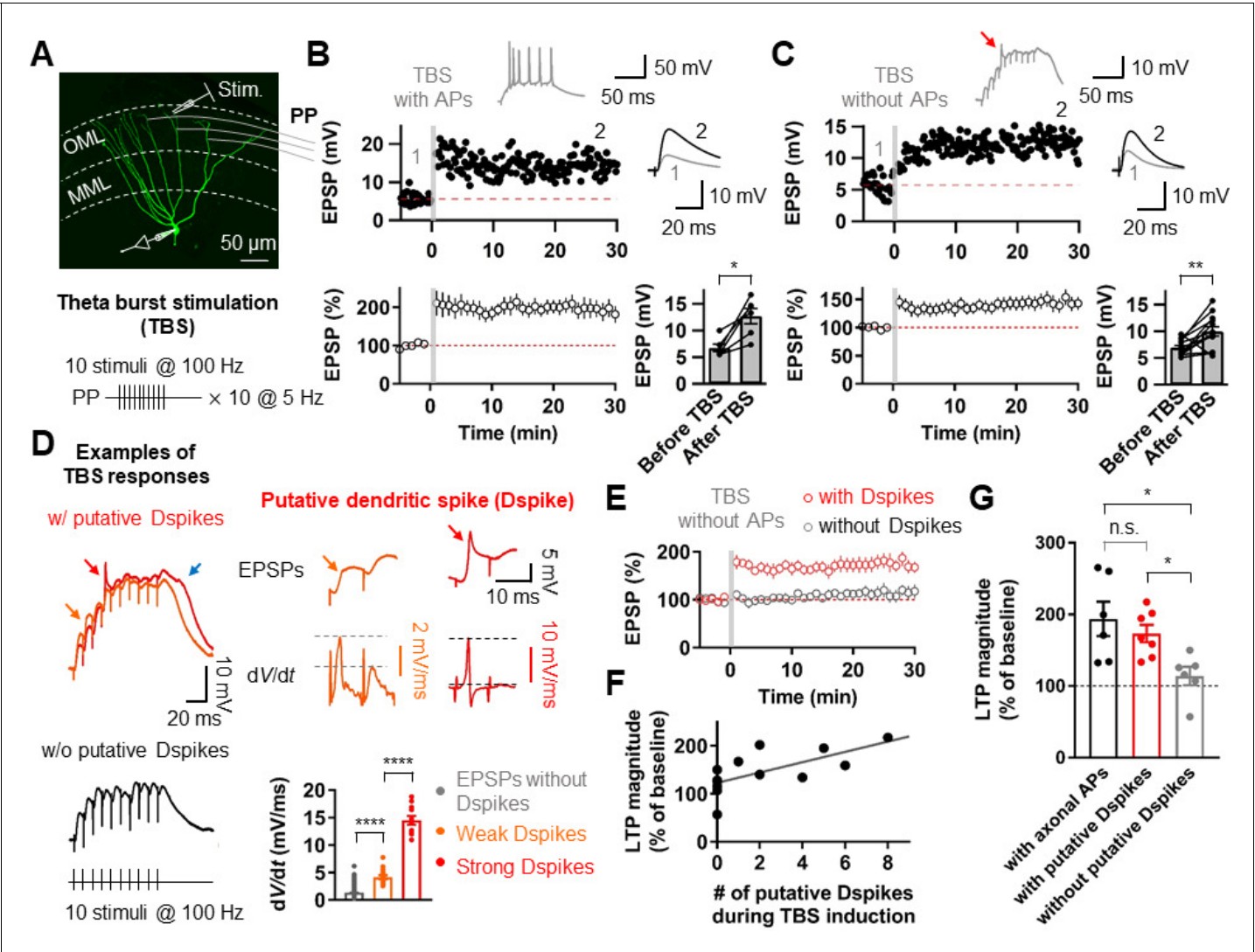

**Figure 1.** Putative dendritic spikes during theta-burst stimulation (TBS) induction are required for long-term potentiation (LTP) at the perforant-path (PP) to granule cell (GC) synapse. (**A**) Maximum intensity projection of confocal stack fluorescence images of a GC (top) indicating the medial molecular layer (MML) and the outer molecular layer (OML). Synaptic responses of the PP were evoked by electrical stimulation in the OML. Scale bar is 50 µm. (bottom) Theta-burst LTP induction protocol. (**B**) Representative time course of EPSPs (top) and summary plot (bottom) before and after TBS of the PP synapses. Red line denotes average EPSP baseline value. Increment in EPSP amplitude denotes LTP. Representative traces, which correspond to the numbers in the time course plot, show the average of 30 EPSP traces before and 25–30 min after TBS (for all subsequent figures). Inset: example of the first burst of TBS responses showing initiation of multiple axosomatic APs during TBS induction upon increasing the stimulus intensity. (bottom, right) Summary bar plot of average EPSP amplitude before and after TBS indicating a significant increment in synaptic responses 25–30 min after TBS stimulation (n = 6, *p<0.05). (**C**) Same as (**B**) but no axosomatic AP initiation occurs during TBS. Inset: representative example of TBS responses. Arrow indicates a putative dendritic spike. (bottom right) Summary plot showing that a significant potentiation in EPSP amplitude was induced after TBS stimulation even in the absence of axosomatic APs (n = 13, **p<0.01). (**D**) (left) Representative somatic voltage traces when TBS stimulation evokes putative dendritic spikes (top) or not (bottom). The arrows indicate weak (orange) and strong (red) putative dendritic spikes and accompanying plateau potentials (blue arrow). (right) Somatic voltages (top row) and corresponding d*V*/d*t* (bottom row) of weak (orange) and strong (red) putative dendritic spikes in the burst responses (left) on an expaned time scale. Note that d*V*/d*t* values of the putative dendritic spikes increase sharply in a non-linear manner. (bottom, right) Summary bar graphs of d*V*/d*t* peak values of somatically recorded EPSPs with or without dendritic spikes (grey: EPSPs without putative Dspikes; orange: with weak putative Dspikes; red: with strong putative Dspikes). (**E**) Average time course of EPSPs when TBS stimulation evokes putative dendritic spikes and in the absence of putative dendritic spikes during TBS (black). LTP is induced only if putative dendritic spikes are present during TBS stimulation. (**F**) The number of putative dendritic spikes are significantly correlated with the magnitude of LTP. Black lines represent linear regressions (n = 13). (**G**) Bar summary graph and individual experiments (circles) indicating that experiments showing the occurrence of putative dendritic spikes (with putative Dspikes, red) during TBS induction induced a significant increment of LTP. Note that there are no significant differences in the magnitude of LTP when TBS stimulation evokes axosomatic APs (black) or putative dendritic spikes (red). Bars indicate mean ± SEM; circles represent data from individual cells. Lines connect data points from the same experiment. *0.01 ≤ P < 0.05. **p<0.01. ****p<0.0001. Single-cell data

*Figure 1 continued on next page*

*Figure 1 continued*
(top panel in B and C) and mean data (bottom panels in B and C, and E; mean ± SEM). Vertical gray bars in B, C, and E indicate the time point of the induction protocol.
DOI: https://doi.org/10.7554/eLife.35269.002
The following source data is available for figure 1:
**Source data 1.** Source data for *Figure 1*.
DOI: https://doi.org/10.7554/eLife.35269.003

*Magee, 2006*; *Kim et al., 2015*), various shapes of putative dendritic spikes were observed during TBS (*Figure 1D*). These voltage responses were first distinguished from EPSPs without dendritic spikes by the spikelet waveform (*Figure 1D*) and then identified as putative dendritic spikes when the peak of the temporal derivative (d$V$/d$t$) of somatic voltage responses was larger than 2.5 mV/ms. Typically, weak (2.5 mV/ms $\leq$ d$V$/d$t$ < 10 mV/ms) and strong (d$V$/d$t$ > 10 mV/ms) putative dendritic spikes were identified (weak dendritic spikes: orange, 4.2 ± 0.4 mV/ms, n = 17; strong dendritic spikes: red, 14.5 ± 0.8 mV/ms, n = 11; p<0.0001; *Figure 1D*). The d$V$/d$t$ of the EPSPs without any putative dendritic spikes was 1.3 ± 0.01 mV/ms (n = 2498), which was significantly smaller than that of weak putative dendritic spikes (p<0.0001; *Figure 1D*). These putative dendritic spikes were accompanied by a sustained plateau potential (*Figure 1D*; see also *Figure 2B*). Remarkably, the presence of these weak and strong putative dendritic spikes during TBS was correlated with LTP induction (in the presence of putative dendritic spikes, 173.2 ± 12.1%, n = 7; in the absence of putative dendritic spikes 114.1 ± 12.8%, n = 6; p<0.005; *Figure 1E,G*). Indeed, we found a strong correlation between the number of putative dendritic spikes observed during TBS and the magnitude of LTP (r = 0.77; p<0.005; n = 13; *Figure 1F*). These results suggest that dendritic spikes but not axosomatic APs contribute to LTP induction at PP-GC synapses.

To further test whether axosomatic spikes contribute to the induction of LTP at the PP-GC synapse, we used associative pairing protocols (*Figure 2*). Presynaptic activity was either followed (pre-postsynaptic sequence) or preceded (post-presynaptic sequence) at a 10 ms interval by single or AP bursts (2 APs at 100 Hz). Both protocols failed to induce significant changes in EPSP amplitude (pre-postsynaptic sequence, control: 4.97 ± 0.38 mV, after induction: 5.83 ± 0.86 mV, n = 15, p=0.23; post-presynaptic sequence, control: 4.78 ± 0.54 mV, after induction: 4.91 ± 0.99 mV, n = 9, p=0.54; *Figure 2A,B*), suggesting that APs of axonal origin are not important for potentiation. We further performed similar experiments at the medial PP–GC synapses by stimulating axons in the middle third of the molecular layer (*Figure 2—figure supplement 1A*). Pairing EPSPs and APs in both pre-postsynaptic and post-presynaptic sequences did not induce LTP of EPSPs (pre-postsynaptic sequence, control: 4.86 ± 0.40 mV, after induction: 5.09 ± 0.99 mV, n = 7, p=0.9015; post-presynaptic sequence, control: 3.86 ± 0.26 mV, after induction: 4.27 ± 0.57 mV, n = 6, p=0.56; *Figure 2—figure supplement 1B–E*), consistent with previous studies that reported no synaptic potentiation after similar pairing protocols (*Yang and Dani, 2014*; *Lopez-Rojas et al., 2016*). Together, these results imply that dendritic spikes, rather than axosomatic APs, are the essential signal for LTP induction at distal GC synapses.

## LTP by TBS at PP–GC synapses requires NMDARs and Na$^+$ channels

We next examined the receptors involved in LTP expression at the PP-GC synapses. Bath application of the NMDAR antagonist DL-AP5 (50 μM) abolished LTP (control: 8.89 ± 0.95 mV; after induction: 9.88 ± 1.21 mV, n = 9, p=0.50; *Figure 3A,E,F*). While sustained plateau potentials during LTP induction were inhibited by DL-AP5 in the external solution, fast rising events remained unchanged (*Figure 3B*). Thus, we next tested whether dendritic voltage-gated Na$^+$ channels are involved in LTP. To block dendritic Na$^+$ channels without affecting synaptic transmission, we included the intracellular sodium channel blocker QX-314 (5 mM) in the whole-cell patch pipette. Stimulus intensity was set to elicit baseline EPSP of similar or slightly larger amplitude than that required to produce putative dendritic spikes during TBS (*Figure 3C,D*). This manipulation abolished both putative dendritic spikes (0 of 7 cells; *Figure 3D*) and TBS-induced LTP (control: 9.24 ± 1.21 mV, after induction: 8.77 ± 1.67 mV, n = 7, p=0.69; *Figure 3C–E*). These results indicate that TBS-induced LTP in GCs depends on the activation of NMDARs and voltage-gated Na$^+$ channels on the postsynaptic

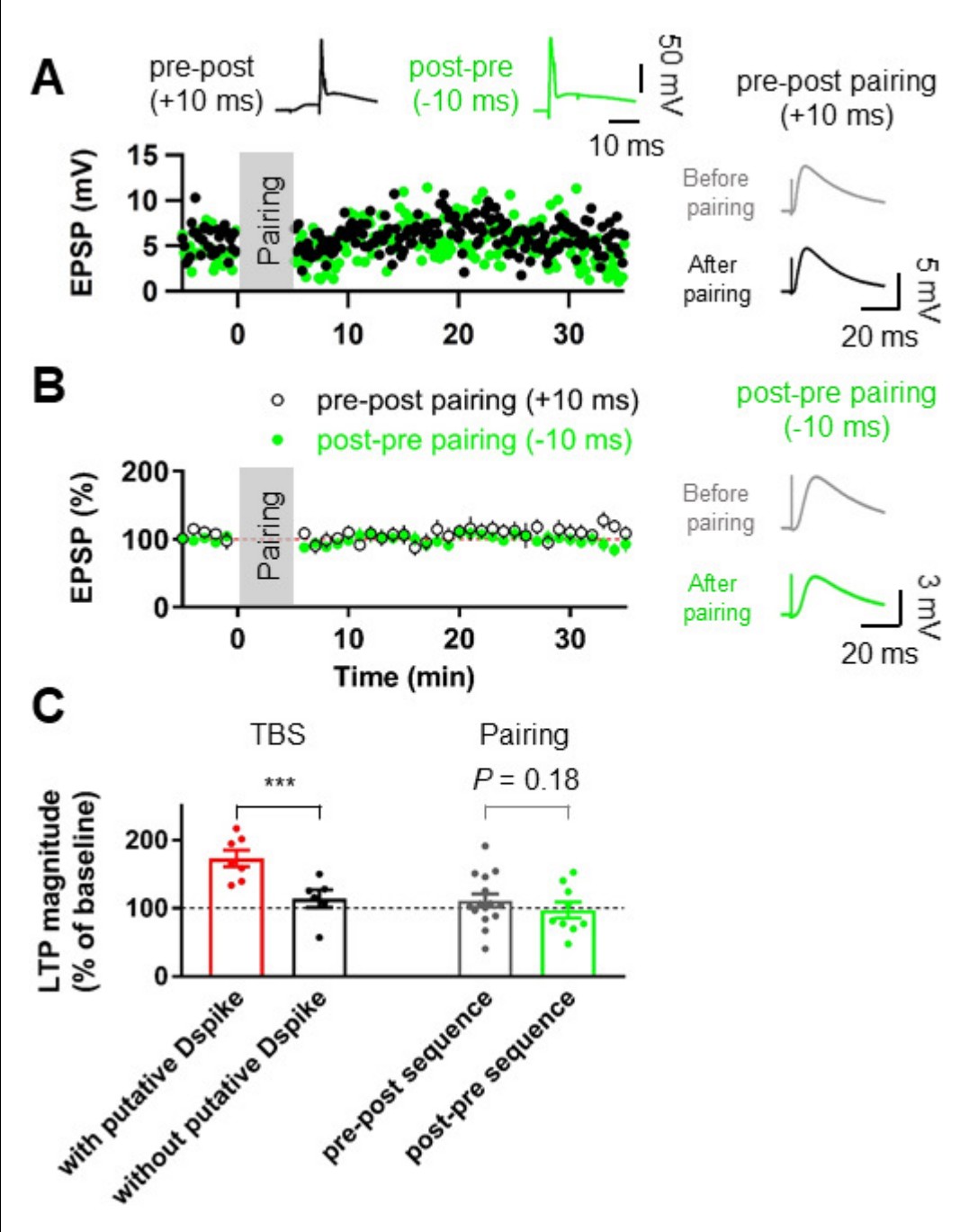

**Figure 2.** Pairing protocols did not induce LTP at the lateral PP-GC synapses. (**A**) Representative time courses of EPSP amplitudes before and after pairing presynaptic stimulation of the PP synapses and postsynaptic action potentials with short time intervals (+10 ms, Pre–post sequence, black, top left inset; –10 ms, Post–pre sequence, green, top right inset). Both pairing protocols induce no significant changes in EPSP, suggesting that action potential (AP) backpropagation is not necessary for LTP induction. Right inset shows that LTP was not induced after pairing of EPSPs and APs in both pre-post (top) and post-pre sequences (bottom). (**B**) The average EPSP time courses of pre-post (black) and post-pre (green) induction protocols of pairings between synaptic responses and postsynaptic APs, showing that low-frequency pairing protocols failed to induce LTP. (**C**) Summary data indicating that experiments showing the occurrence of putative dendritic spikes during TBS protocol induced a significant increment of LTP, whereas pairings of synaptic stimulation with postsynaptic APs did not show a statistically significant LTP induction, independent of temporal order. Bars indicate mean ± SEM; circles represent data from

*Figure 2 continued*

individual cells. Lines connect data points from the same experiment. ***p<0.005. Single-cell data (**A**) and mean data (**B**); mean ± SEM. Vertical gray bars in (**A**) and (**B**) indicate the time point of the induction protocol.
DOI: https://doi.org/10.7554/eLife.35269.004

The following source data and figure supplements are available for figure 2:

**Source data 1.** Source data for *Figure 2*.
DOI: https://doi.org/10.7554/eLife.35269.006

**Figure supplement 1.** Pairing protocols did not induce LTP at the medial perforant path (MPP)-GC synapses.
DOI: https://doi.org/10.7554/eLife.35269.005

**Figure supplement 1—source data 1.** Source data for *Figure 2—figure supplement 1*.
DOI: https://doi.org/10.7554/eLife.35269.007

dendritic membrane, reinforcing the idea that dendritic Na⁺ spikes may play a pivotal role in TBS induction at distal synapses.

## Backpropagation of axosomatic APs in the dendrites of GCs

To directly dissect the dendritic mechanism that determines the induction rules of synaptic plasticity in GCs, we employed subcellular patch-clamp techniques to analyze AP backpropagation and initiation in GC dendrites (*Figure 4A*). First, we characterized backpropagation in GCs with somatic current injection evoking trains of APs at the soma while simultaneously recording in the dendrite (*Figure 4B*). The AP always appeared first in the somatic recording and then in the dendrites (*Figure 4C*). Similar to the previous report (*Krueppel et al., 2011*), the peak amplitude of the bAPs attenuated as a function of distance from the soma (*Figure 4D,E*). At dendritic distances beyond 150 µm from the soma, the peak amplitude of bAPs was reduced to ~36% of the amplitude of somatic APs (soma: 93.6 ± 1.7 mV; dendrite: 33.7 ± 3.1 mV; n = 10, p<0.005; *Figure 4D*). The attenuation per dendritic length is much more pronounced in GCs compared to the other neocortical (*Stuart et al., 1997*; *Nevian et al., 2007*) and hippocampal pyramidal neurons (*Spruston et al., 1995*; *Kim et al., 2012*). It is interesting to note that the extent of bAP attenuation when normalized to the overall length of GC dendrites (mean dendritic length: 278 ± 7.4 µm; n = 11) is quite similar to layer five pyramidal neuron dendrites (cf. Supplementary Figure 3B in *Nevian et al. (2007)*; see also *Krueppel et al. (2011)*; *Figure 4F*), suggesting that the failure of LTP induction during pairing protocols at distal GC synapses (*Figure 2*) might be explained by the voltage attenuation of bAPs, as was reported in layer five pyramidal neurons (*Letzkus et al., 2006*; *Sjöström et al., 2008*); Feldman; 2012). We also estimated the conduction velocity of bAPs by analyzing the AP latencies measured at the half-maximal amplitude of the rising phase. APs were initiated axonally and propagated back into the dendrites with a velocity of 226 µm/ms (*Figure 4G*; n = 56; corresponding to 0.2–0.3 m/s, *Senzai and Buzsáki, 2017*). This conduction velocity is slower than those measured in the apical and basal dendrites of layer five pyramidal neurons (apical dendrites, 508 µm/ms; basal dendrites, 341 µm/ms; *Nevian et al. (2007)* and *Stuart et al., 1997*) and is similar or lower than the velocity estimated in the apical dendrite of other hippocampal principal neurons (*Spruston et al., 1995*; *Kim et al., 2012*). In summary, these experiments show that bAPs in GCs propagate into the dendrites with substantial voltage attenuation and moderate conduction velocity.

## Ionic mechanisms of AP backpropagation

To determine the ionic mechanisms underlying the strong voltage attenuation and the moderate conduction velocity of bAPs, we assayed the somatic and dendritic distribution of voltage-gated Na⁺ and K⁺ currents in outside-out patches isolated at various locations using pipettes of similar open tip resistance (soma: 17.9 ± 0.5 MΩ, n = 24; dendrite: 19.4 ± 0.5 MΩ, n = 36; *Figure 5*). Depolarizing voltage pulses from –120 mV to 0 mV evoked TTX-sensitive inward Na⁺ currents in the majority of outside-out patches excised from both soma and dendrites (*Figure 5A* and *Figure 5—figure supplements 1* and *2*). Pooled data demonstrated that the current amplitude of Na⁺ channels is uniformly distributed over the dendritic membrane (*Figure 5D*). On average, the peak amplitude of Na⁺ currents was not significantly different between somatic and dendritic patches (soma: –6.81 ± 1.36 pA, n = 19; proximal dendrite (within 100 µm, PD): –5.02 ± 1.55 pA, n = 8; distal dendrite

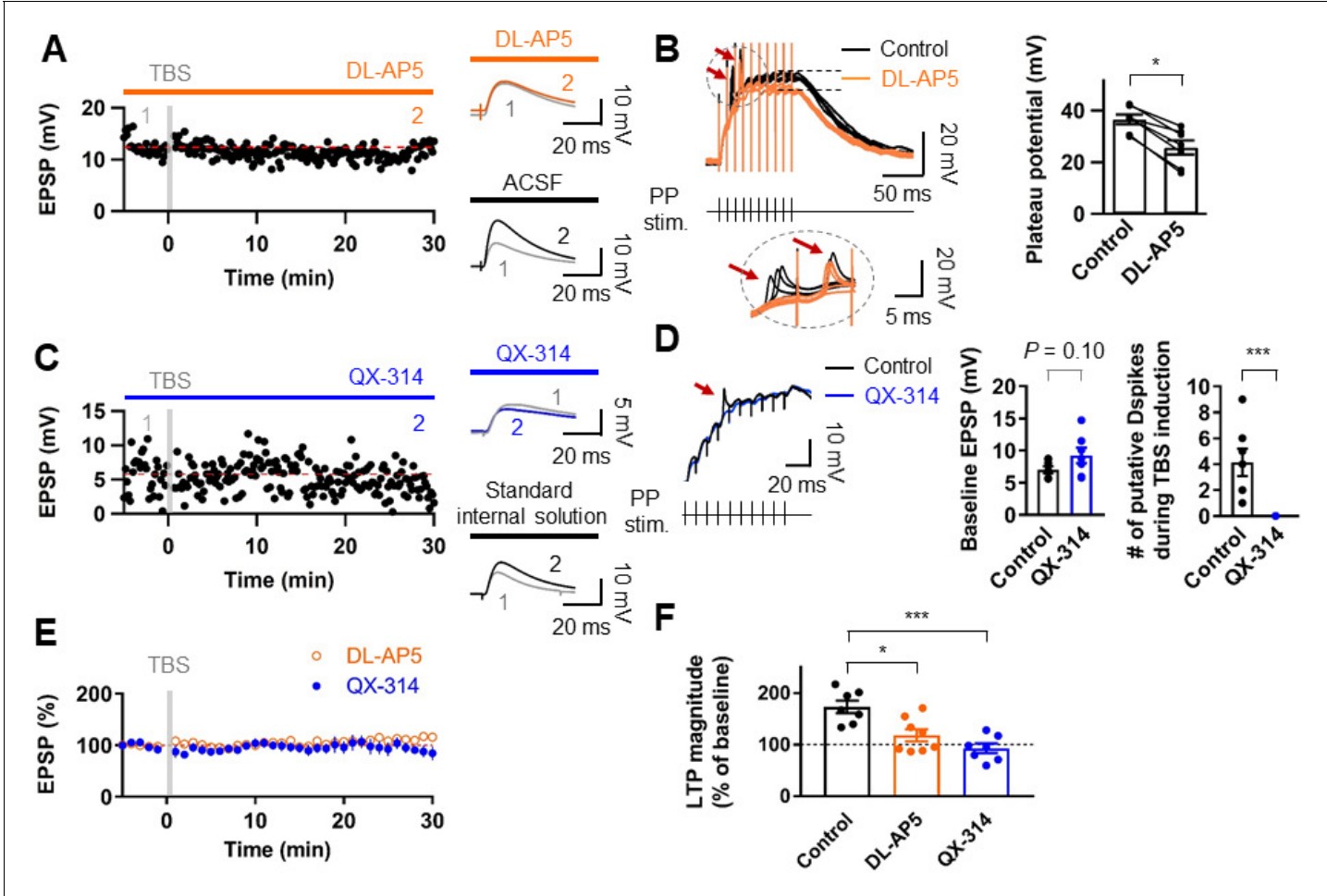

**Figure 3.** Induction of LTP at PP–GC synapses requires activation of NMDARs and the involvement of Na$^+$ channels. (A) An example of the time course of EPSP amplitude when TBS protocols was applied in the presence of the NMDAR antagonist, DL-AP5 (50 µM). Red line indicates the average EPSP amplitude. Insets show that EPSP did not increase in the presence of DL-AP5 (orange) after TBS, but a robust LTP was induced when TBS was applied in standard saline (ACSF, black). (B) (left) Long-duration plateau potentials are mediated by NMDA receptor channels. Somatically recorded voltages in response to high-frequency burst sitmulation (green, 10 shocks, 100 Hz) under control (black) and in the presence of DL-AP5 (50 µM; orange). The inset shows putative dendritic spikes (indicated by the red arrow) before and after the addition of DL-AP5; Note that putative dendritic spikes are resistant to the NMDAR blockers. (right) Summary of the effects of DL-AP5 on plateau potentials. Peak amplitdue of plateau potentials were measured after the stimulus (indicated by dashed lines; Control: 36.6 ± 2.2 mV; DL-AP5: 25.6 ± 3.2 mV; n = 6, *p<0.05). (C) A representative time course of EPSP amplitude before and after TBS when the cells were dialyzed with a sodium-channel blocker, QX-314 (5 mM). Inset shows that the averaged EPSP amplitude did not change when blocking sodium channels with QX-314 (blue) despite TBS induction. In contrast, when cells were dialyzed with the standard intracellular solution, the amplitude of EPSP increased after TBS (i.e. LTP). (D) The effects of intracellular QX-314 on dendritic spike initiation in response to high-frequency PP sitmulation. (left) Representative traces of EPSPs in response to PP stimulation with (blue) and without (black, control) QX-314. Putative dendritic spikes (arrow) were observed only under control condition. (right) Bar graphs indicate the baseline EPSP amplitude (EPSP$_{control}$: 7.05 ± 0.54 mV, n = 7; EPSP$_{QX-314}$: 9.24 ± 1.21 mV, n = 7; p=0.10) and the number of putative dendritic spikes during TBS induction (control: 4.14 ± 1.06, n = 7; QX-314: 0, n = 7; ***p<0.005) in two groups. (E) Summary plot of TBS-induced LTP experiments in the presence of DL-AP5 (orange) and with dialysis of QX-314 in the recording pipette (blue). Both treatments prevented the induction of LTP at the PP to GC synapse. (F) Summary bar graph and individual average EPSP amplitudes after TBS in control (standard saline, black), bath application of DL-AP5 (orange) and dialysis of QX-314 (blue). Treatments with DL-AP5 and QX-314 showed a significant difference compared to the standard LTP induction (DL-AP5, *p<0.05; QX-314, ***p<0.005; compared to control in *Figure 1I*). Representative traces in A and C correspond to the numbers (1 and 2) n the time-course plot. Bars indicate mean ± SEM; circles represent data from individual cells. Lines connect data points from the same experiment. Single-cell data (A,C) and mean data (E; mean ± SEM). Vertical gray bars in (A, C,) and (E) indicate the time point of the induction protocol.

DOI: https://doi.org/10.7554/eLife.35269.008
The following source data is available for figure 3:

**Source data 1.** Source data for *Figure 3*.
DOI: https://doi.org/10.7554/eLife.35269.009

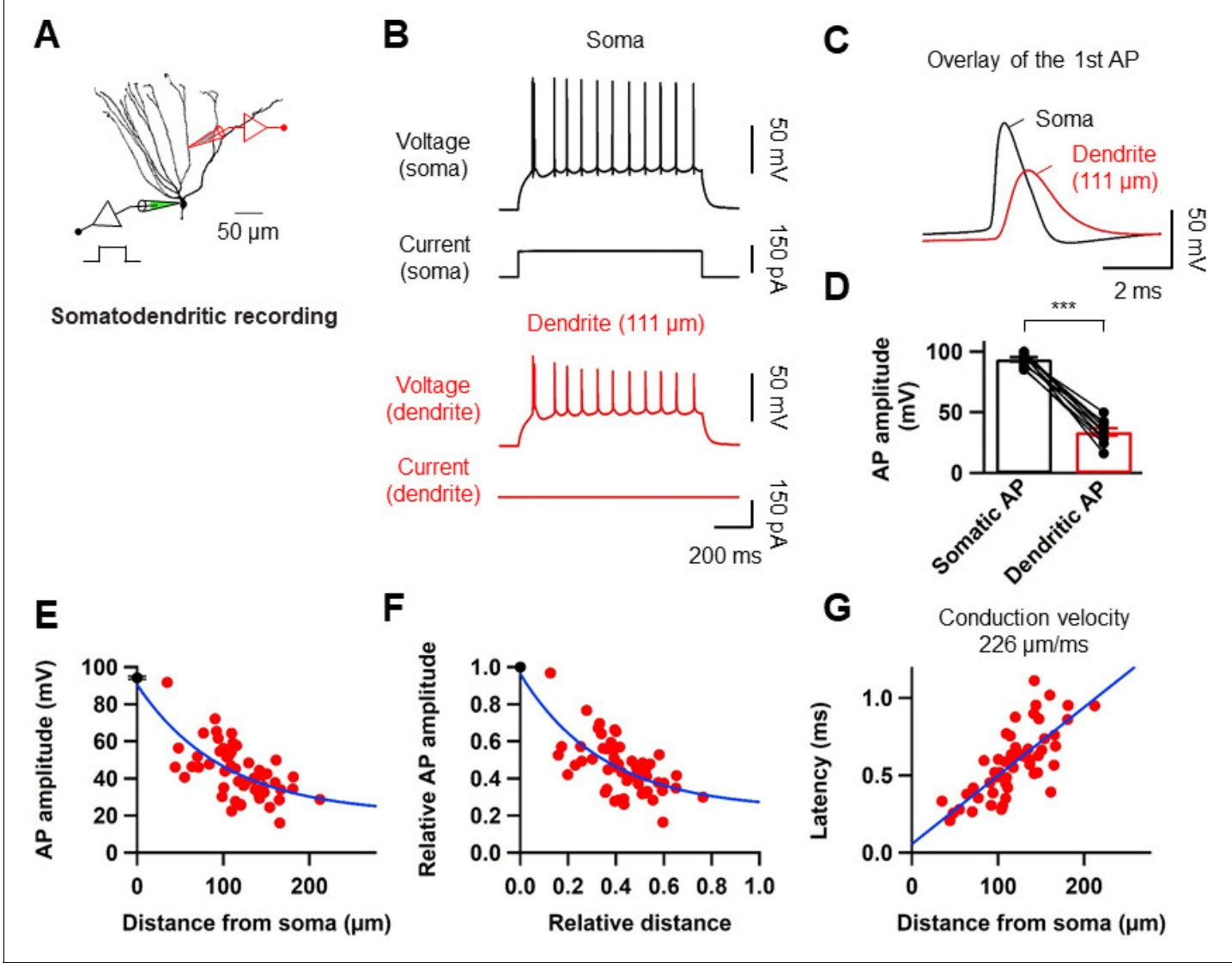

**Figure 4.** Properties of backpropagating APs in the dendrites of GCs. (A) Morphological reconstruction of a GC with representative double somatic and dendritic whole-cell recording configuration used to analyze the AP backpropagation. Scale bar is 50 µm. (B) A train of APs elicited by a 1 s current pulse applied at the soma. Black traces indicate somatic voltage and corresponding current; red traces indicate dendritic voltage and corresponding current. (C) First AP in the train displayed at expanded time scale. Voltage traces (soma in black, dendrite in red) indicate that the AP is initiated first near the soma and propagated back into the dendrites with a lower amplitude. (D) Summary graph to compare somatic (black) and dendritic (red) AP peak amplitude. Bars indicate mean ± SEM; circles represent data from individual cells. Lines connect data points from the same experiment. ***p<0.005. (E) Scatter plot of peak amplitude of the backpropagating AP against the absolute physical distance of the recording site from the soma (56 somatodendritic recordings). The blue curve represents a mono-exponential fit to the data points between 0 and 212 µm. (F) Scatter plot of the bAP amplitude normalized to the corresponding axosomatic AP amplitude plotted against the distance of the recording site scaled to the total dendritic length (278 ± 7.4 µm; n = 11). The blue curve is a mono-exponential fit to the data. (G) Scatter plot of AP latency as a function of the distance from the soma (56 somatodendritic recordings) together with a linear regression (blue line) to compute the average conduction velocity of the AP into the dendrites; dendritic AP propagation velocity was 226 µm/ms. Single-cell data (E–G, red) and mean data (E), black; mean ± SEM).

DOI: https://doi.org/10.7554/eLife.35269.010

The following source data is available for figure 4:

**Source data 1.** Source data for *Figure 4*.
DOI: https://doi.org/10.7554/eLife.35269.011

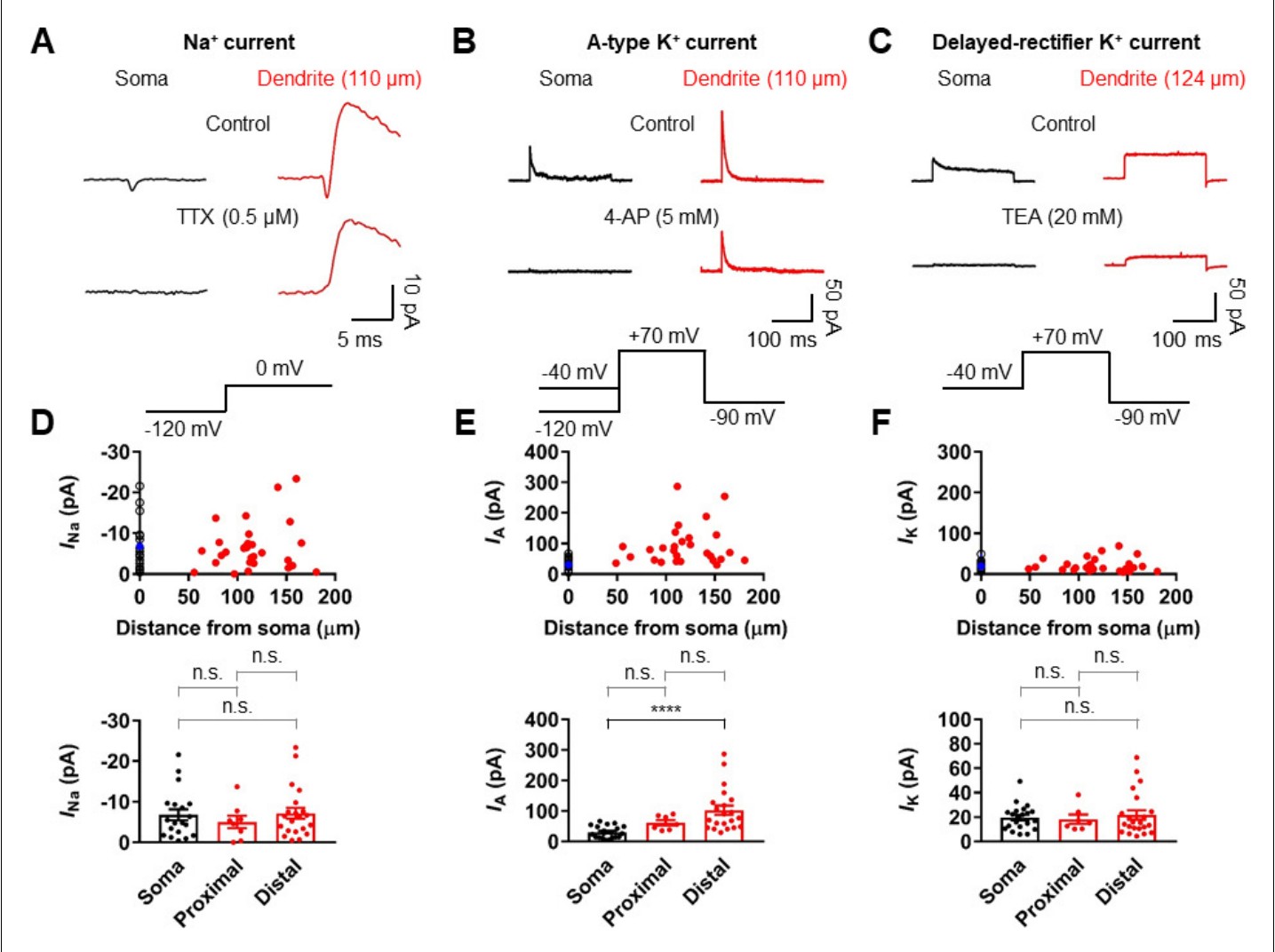

**Figure 5.** Differential Na$^+$ and K$^+$ channel densities in the dendrites of GCs. (A) Averages of Na$^+$ current recorded from outside-out patches from soma (black, averages of 25–27 sweeps) and dendrite (red, 110 µm, averages of 20 sweeps) in response to a test pulse potential to 0 mV (bottom). Na$^+$ currents were recorded in the presence of 4-AP (5 mM) and TEA (20 mM). Left, soma; right, dendrite; Top, control; bottom, currents in the presence of 0.5 µM TTX in the bath. Leak and capacitive currents were subtracted by a 'P over –4' correction procedure. Note that the remaining outward current is the resistant K$^+$ current component to 5 mM 4-AP (**Figure 5—figure supplements 1** and **2**; **Hoffman et al., 1997**). Blockade of outward K$^+$ currents by extracellular 4-AP (5 mM) only had a negligible effects on the peak amplitude of Na$^+$ currents (**Figure 5—figure supplement 2**). (B) Averages of A-type K$^+$ current evoked in outside-out patches excised from soma (black, averages of 6–8 sweeps) and dendrite (red, 110 µm averages of 15–19 sweeps) in response to a test pulse potential to +70 mV (top). Transient A-current was measured by subtraction of traces with a −40 mV prepulse from those with a −120 mV prepulse. Left column, soma; Right column, dendrite; Top row, control; Bottom row, currents in the presence of 5 mM 4-AP in the bath. (C) Averages of delayed rectifier K$^+$ current evoked in outside-out patches excised from soma (black, averages of 6–8 sweeps) and dendrite (red, 124 µm, averages of 10–18 sweeps) in response to a test pulse potential to +70 mV (top). Delayed rectifier K$^+$ current was measured by a −40 mV prepulse. Left column, soma; Right column, dendrite; Top row, control; Bottom row, currents in the presence of 20 mM TEA in the bath. See also **Figure 5—figure supplement 1**. (D, E, F) (top) Plot of amplitude of Na$^+$ channel activity (D), A-type K$^+$ channel activity (E), and delayed rectifier K$^+$ channel activity (F) as a function of distance from the soma, demonstrating that various channels are differentially expressed across the length of GC dendrites. Data from 19, 21, and 21 somatic (black circles) and 29, 28, and 29 dendritic patches (red circles). Blue circles represent the average of somatic recordings. (bottom) Summary bar graph showing the peak amplitude of Na$^+$ (D), A-type K$^+$ (E) and delayed rectifier K$^+$ (F) channel activity in the soma, proximal dendrite (<100 µm) and distal dendrite (≥100 µm). Bars indicate mean ± SEM; circles represent data from individual experiments. n.s., not significant; ****p<0.0001 by Kruskal Wallis test with post hoc multiple comparison using Dunn's test.

DOI: https://doi.org/10.7554/eLife.35269.012

The following source data and figure supplements are available for figure 5:

**Source data 1.** Source data for **Figure 5**.

DOI: https://doi.org/10.7554/eLife.35269.016

*Figure 5 continued on next page*

*Figure 5 continued*

**Figure supplement 1.** Pharmacological analysis of voltage-dependent Na$^+$ and K$^+$ currents.

DOI: https://doi.org/10.7554/eLife.35269.013

**Figure supplement 1—source data 1.** Source data for *Figure 5—figure supplement 1*.

DOI: https://doi.org/10.7554/eLife.35269.017

**Figure supplement 2.** The dose-dependent effect of 4-AP on transient outward currents and Na$^+$ currents.

DOI: https://doi.org/10.7554/eLife.35269.014

**Figure supplement 2—source data 1.** Source data for *Figure 5—figure supplement 2*.

DOI: https://doi.org/10.7554/eLife.35269.018

**Figure supplement 3.** Effect of 4-AP on AP backpropagation.

DOI: https://doi.org/10.7554/eLife.35269.015

**Figure supplement 3—source data 1.** Source data for *Figure 5—figure supplement 3*.

DOI: https://doi.org/10.7554/eLife.35269.019

(beyond 100 µm, DD): –7.17 ± 1.36 pA, n = 21; p>0.99 for all cases, Kruskal-Wallis test with Dunn's multiple comparison; *Figure 5D*). Surprisingly, very large transient K$^+$ currents that appeared to activate and inactivate rapidly in response to voltage pulses from –120 mV to +70 mV was found in patches obtained from the dendrites (*Figure 5B*). These inactivating components of K$^+$ channel activity were reduced by the A-type K$^+$ channel blocker 4-aminopyridine (4-AP, *Figure 5B* and *Figure 5—figure supplements 1* and *2*). Plotting the current amplitude along the dendrite demonstrated that GC dendrites show significantly larger A-type K$^+$ currents than the soma (soma: 30.4 ± 4.2 pA, n = 21; PD: 61.5 ± 8.7 pA, n = 7; DD: 102.3 ± 15.3 pA, n = 21; soma vs. PD, p=0.07; soma vs. DD, p<0.0001; PD vs. DD, p=0.96, Kruskal-Wallis test with Dunn's multiple comparisons; *Figure 5E*). Consistent with this finding, we found that bath application of 5 mM 4-AP in simultaneous somatic and dendritic voltage recordings results in a significant enhancement of the peak amplitude and duration of bAPs in the dendrites, indicating that dendritic A-type K$^+$ current (I$_A$) contributes to voltage attenuation of bAPs in GCs (*Figure 5—figure supplement 3*). Finally, the delayed rectifier K$^+$ current components, which was largely inhibited by 20 mM extracellular tetraethylammonium (TEA, *Figure 5C* and *Figure 5—figure supplement 1*), showed a low and uniform expression over the entire somatodendritic axis (soma: 19.6 ± 2.2 pA, n = 21; PD: 18.3 ± 3.8 pA, n = 7; DD: 21.7 ± 3.8 pA, n = 22; p>0.99 for all cases, Kruskal-Wallis test with Dunn's multiple comparisons; *Figure 5C,F*). Taken together, these voltage-clamp data reveal that a markedly high density of K$^+$ channels and a uniform density of Na$^+$ channels are present in the dendrites of GCs, which are likely to account for both the strong dendritic AP attenuation and the moderate AP propagation velocity.

## A burst of PP stimuli can evoke dendritically initiated Na$^+$ spikes in GCs

Voltage-gated Na$^+$ channels may contribute to the generation of dendritically initiated local spikes in regions where the dendrites are very small (*Holmes, 1989*). Indeed, depolarizing dendrites with brief current pulse injection evoked local spikes in distal dendrites of GCs (55 of 63 recordings; *Figure 6A–C*). In the proximal domain (within 100 µm from the soma), dendritic spikes were not detected and depolarizing dendrites resulted in an axosomatic AP that propagated back to the dendritic recording site (*Figure 6D*). At dendritic recording locations 70 to 150 µm from the soma, dendritic spikes were associated with axosomatic spikes. For distances larger than 150 µm from the soma (approximately corresponding to the outer molecular layer), current injections robustly initiated isolated dendritic spikes (*Figure 6D*). Pharmacological analysis revealed that dendritic spikes were resistant to 200 µM CdCl$_2$ and 50 µM NiCl$_2$ but were inhibited by 0.5 µM TTX, indicating that they were mediated by dendritic voltage-gated Na$^+$ channels rather than by Ca$^{2+}$ channels (TTX: 11.3 ± 3.3%, n = 3; CdCl$_2$: 93.9 ± 7.8%, n = 4; NiCl$_2$: 97.9 ± 10.2%, n = 4; *Figure 6—figure supplement 1*). Because our results indicate that GC dendrites contain a high density of transient, A-type K$^+$ channels (*Figure 5B,E*), we further explored how these channels influence dendritic spike initiation. We combined dual somatodendritic recordings and focal application of 4-AP (10 mM) to the dendritic patch (*Figure 6—figure supplement 2A*). Local block of I$_A$ significantly decreased the current threshold for dendritic spike initiation, suggesting that dendritic I$_A$ affect the generation of dendritic spikes (*Figure 6—figure supplement 2B,C*).

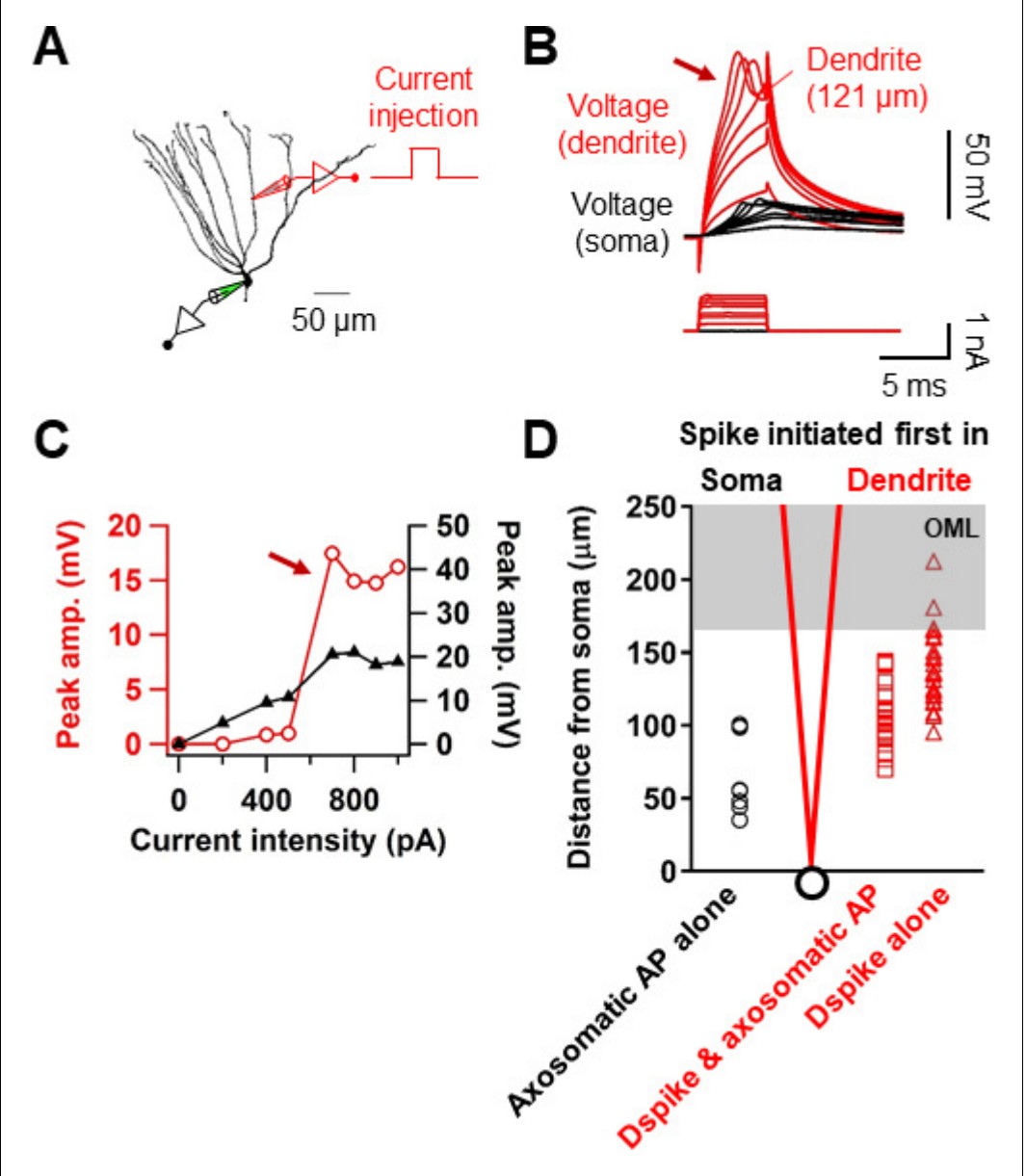

**Figure 6.** Initiation of dendritic spikes in GCs. (**A**) Schematic recording configuration of a simultaneous somatic (black) and dendritic (red) patch-clamp recording on a GC. (**B**) Local dendritic spikes (arrow) in GCs evoked by dendritic current injection pulses of increasing amplitude (black, voltage in the soma; red, voltage in the dendrite). (**C**) The relationship between voltage amplitude (black, soma; red, dendrite) and dendritic injection resembles a step-function, suggesting the all-or-none nature of the dendritic spike (arrow). Peak values of dendritic spikes were measured after subtraction of scaled subthreshold dendritic responses. (**D**) A summary plot showing whether a spike was evoked first in the dendrite (right, red) or in the soma (left, black) for an increasing current pulse injection at the dendrite. Red squares indicate the cells showing a dendritic spike followed by an axosomatic spike. Red triangles show the cells showing isolated dendritic spike. Black circles indicate the cells showing only an axosomatic spike. The green shaded area approximately corresponds to the outer molecular layer. The black circle and red lines in the plot indicate the soma and the dendrite, respectively.

DOI: https://doi.org/10.7554/eLife.35269.020

The following source data and figure supplements are available for figure 6:

**Figure supplement 1.** Dendritic spikes are mediated by voltage-gated Na$^+$ channels.
DOI: https://doi.org/10.7554/eLife.35269.021

**Figure supplement 1—source data 1.** Source data for *Figure 6—figure supplement 1*.

*Figure 6 continued on next page*

*Figure 6 continued*

DOI: https://doi.org/10.7554/eLife.35269.023

**Figure supplement 2.** Dendritic A-type K$^+$ channels control dendritic spike initiation.

DOI: https://doi.org/10.7554/eLife.35269.022

**Figure supplement 2—source data 1.** Source data for *Figure 6—figure supplement 2*.

DOI: https://doi.org/10.7554/eLife.35269.024

We next examined whether Na$^+$ spikes in GC dendrites can be evoked by synaptic stimulation of PP inputs. Extracellular synaptic stimulation was combined with simultaneous double patch-clamp recordings from the soma and dendrites (*Figure 7A*). Dendritic spikes could be triggered by stimulating the PP inputs with TBS protocol (*Figure 7A,B*). Repeated trials of a high-frequency PP stimulation using the same stimulus intensity produced variable dendritic responses with subthreshold EPSP, weak dendritic spikes, or strong dendritic spikes that appeared either in isolation or associated with axosomatic APs (*Figure 7B*). When dendritic spikes were present, the d$V$/d$t$ of the corresponding somatic voltages showed variable peak amplitudes that were similar to those of the putative dendritic spikes during TBS induction shown in *Figure 1D* (EPSPs with a dendritic spike: d$V$/d$t$ = 7.1 ± 1.1 mV/ms, n = 26; p=0.58 compared to EPSPs with putative dendritic spikes: d$V$/d$t$ = 8.2 ± 1.0 mV/ms, n = 28 in *Figure 1D*; *Figure 7C,D*). Overall, dendritic recordings reveals that a high-frequency burst activation of PP synapses can produce dendritic Na$^+$ spikes that are manifested as rapidly rising voltage responses at the soma (*Figure 1D*).

## Dendritically initiated Na$^+$ spikes are required for TBS-induced LTP at PP–GC synapses

Finally, we examined whether dendritic Na$^+$ spikes are necessary for TBS-induced LTP at PP synapses. To this end, we applied a low concentration of TTX (10 nM; *Kim et al., 2015*) to block dendritic Na$^+$ spikes without interfering synaptic transmission (*Figure 8A,B*). Inhibition of dendritic Na$^+$ spikes prevented TBS-induced LTP (control: 6.58 ± 0.51 mV; after induction: 7.04 ± 0.88 mV, n = 8; p=0.64; *Figure 8C–E*), without affecting the baseline EPSP amplitude (control: 5.21 ± 0.78 mV; after TTX: 5.06 ± 0.68 mV, n = 11; p=0.83; *Figure 8—figure supplement 1*). All together, these results suggest that dendritically generated local Na$^+$ spikes in response to TBS of the PP are necessary for LTP induction in GCs.

## Discussion

In summary, the present study demonstrates several major findings. First, our results show that conventional STDP protocols (*Dan and Poo, 2006*; *Feldman, 2012*) do not trigger synaptic potentiation at PP-GC synapses. Second, a physiologically relevant TBS paradigm can efficiently induce LTP in GCs without axosomatic APs. Finally, direct dendritic recordings revealed that LTP induction requires dendritic spikes. To our knowledge, these studies show the first direct evidence for dendritic spike generation in GCs and the role of these spikes during induction of LTP in the dentate gyrus network. Whether our findings also hold for GCs at early stages of development remains to be determined.

Considerable evidence has shown that the electrical properties of neurons at the cellular and subcellular levels are brain region-specific and cell-type-specific and endow unique rules and characteristics on circuit function (*Sjöström et al., 2008*; *Stuart and Spruston, 2015*). Hyperpolarized resting potential, strong dendritic voltage attenuation, and low occurrence of APs are the known electrophysiological characteristics of mature GCs (*Scharfman and Schwartzkroin, 1990*; *Schmidt-Hieber et al., 2004*; *Mongiat et al., 2009*; *Krueppel et al., 2011*; *Pernía-Andrade and Jonas, 2014*), which are distinct from other types of hippocampal principal neurons (*Spruston et al., 1995*; *Kim et al., 2012*; *Stuart and Spruston, 2015*). While backpropagated APs are an important associative signal for triggering plasticity at the synaptic site via the voltage-dependent relief of Mg$^{2+}$ block of NMDARs (*Magee and Johnston, 1997*; *Dan and Poo, 2006*; *Feldman, 2012*; *Mishra et al., 2016*), the above features of mature GCs are highly unfavorable for LTP induction at distal synaptic contacts. Accordingly, we found that GCs do not show LTP at PP-GC synapses during standard

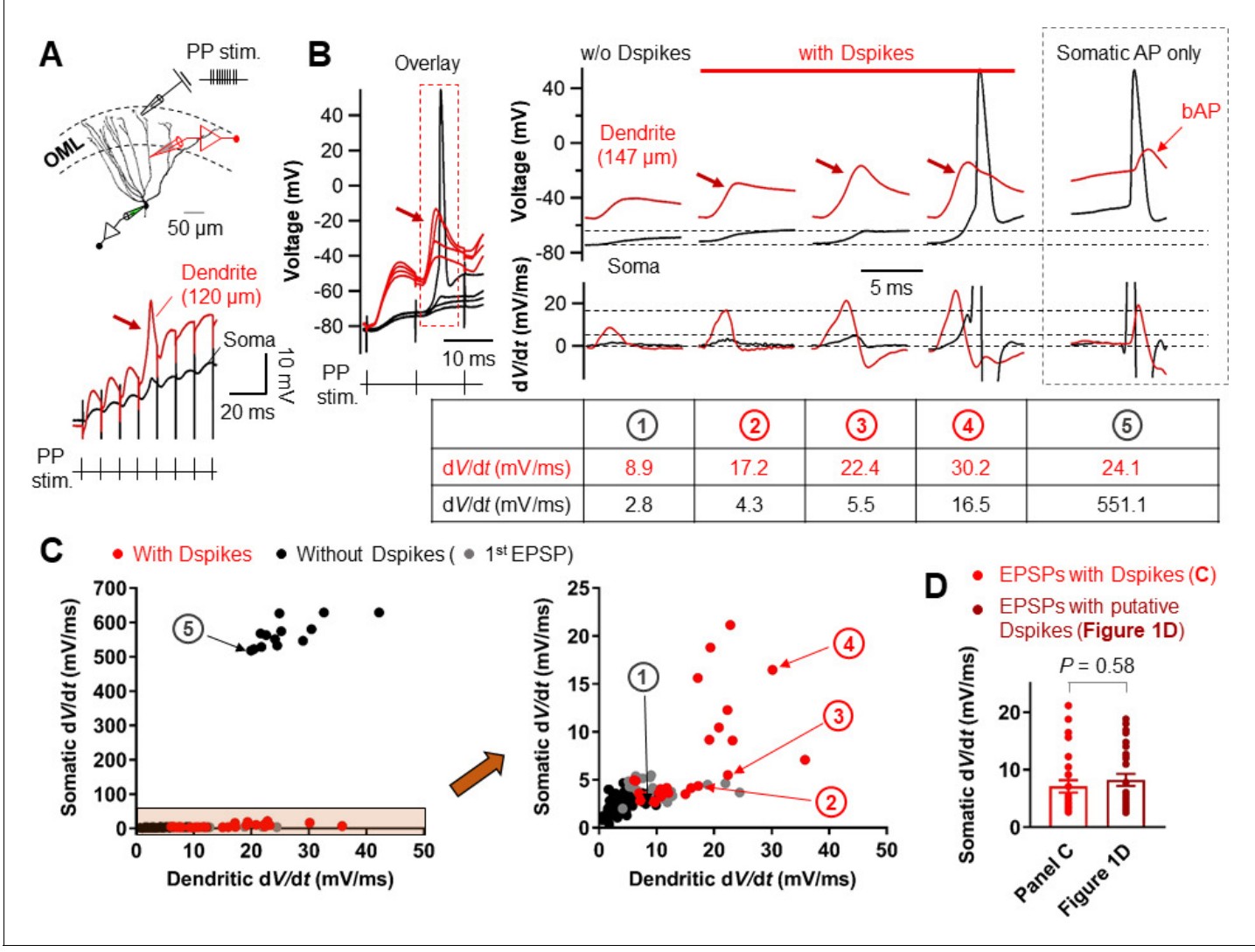

**Figure 7.** Relationship between the d$V$/d$t$ of somatically and dendritically recorded voltages during dendritic spikes generation. (A) (top) Schematic recording configuration of a triple pipette consisting of electrical stimulation of the PP synapses in the OML, dendritic patch-clamp recording (120 µm from the soma, red) and somatic whole-cell recording (black). Scale bar is 50 µm. (bottom) Dendritic spikes can be identified as larger spikes in the dendrite (red arrow) with the corresponding small spike at the soma (black). (B) (left) Somatic and dendritic voltages in response to a high-frequency PP stimulation with a constant stimulus intensity are shown superimposed for comparison. The recording site on the dendrite is 147 µm from the soma. (right) Traces of somatic (black) and dendritic (red) voltage responses (top row) on the left (red box) and corresponding d$V$/d$t$ (bottom). d$V$/d$t$ peak amplitudes of each traces were summarized in the table (bottom). The red arrows indicate the dendritic spikes. When dendritic spikes were present, the corresponding somatic voltage changes were used for analysis of d$V$/d$t$ peaks. For comparison, somatic and dendritic membrane voltages and corresponding d$V$/d$t$ traces during axosomatic AP generation are also shown on the right (dashed box). Encircled numbers indicate correspondence between traces in B) and data points in C). (C) (left) Summary plot of peaks in somatic d$V$/d$t$ against corresponding peaks in dendritic d$V$/d$t$ for four simultaneous somatodendritic recordings in response to high-frequency burst stimulation of the PP synapses (dendritic recording sites are from 136 µm to 175 µm from the soma). (right) An enlarged view of the box (orange) in the left panel. Black circles, in the absence of dendritic spikes; red circles, in the presence of the dendritic spikes; gray circles, data points from the first EPSPs (EPSP$_1$). Note that somatic d$V$/d$t$ peak amplitudes of subthreshold EPSP1s were comparable to those of the somatic traces when dendritic spikes were present. (D) Summary bar graphs of d$V$/d$t$ peak amplitudes of somatically recorded dendritic spikes in C) (red) and in **Figure 1D**. Bars indicate mean ± SEM; circles represent data from individual cells.

DOI: https://doi.org/10.7554/eLife.35269.025

The following source data is available for figure 7:

**Source data 1.** Source data for **Figure 7**.
DOI: https://doi.org/10.7554/eLife.35269.026

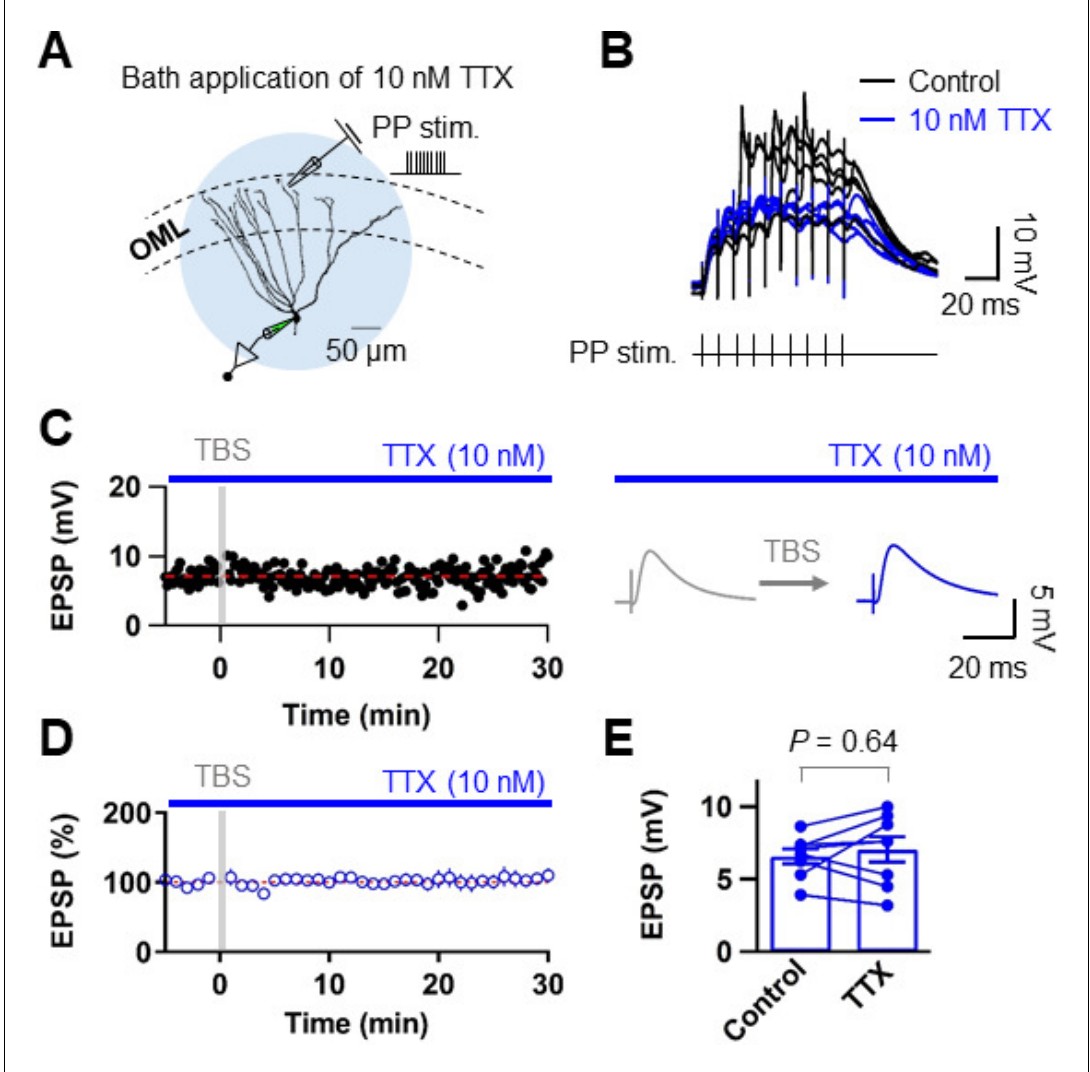

**Figure 8.** Blockade of Na$^+$ channels prevent dendritic spikes and LTP induction by TBS. (A) Schematic diagram showing the experimental configuration. In these experiments, the sodium channel blocker TTX was applied by bath perfusion. Scale bar is 50 μm. (B) Bath application of TTX (10 nM) prevents the generation of dendritic spikes evoked by synaptic stimulation (black), as seen on the somatic whole-recording trace (blue). Note that decreasing availability of Na$^+$ channels by 10 nM TTX had a negligible effect on the EPSP$_1$ (see also *Figure 8—figure supplement 1*). (C) Time course of EPSP amplitudes in the presence of TTX (10 nM). The induction of LTP by TBS can be prevented by bath application of TTX, since the average EPSP amplitude does not change. (D) Average EPSP before and after TBS shows no changes in amplitude in the presence of TTX. (E) Summary bar graph and single experiments (circles) indicating that the TTX application does not produce an increment in the EPSP amplitude and therefore no statistically significant LTP induction. Bars indicate mean ± SEM; circles represent data from individual cells. Data points from the same experiment are connected by lines. Single-cell data (C) and mean data (D); mean ± SEM. Vertical gray bars in C) and D) indicate the time point of the induction protocol.

DOI: https://doi.org/10.7554/eLife.35269.027

The following source data and figure supplements are available for figure 8:

**Source data 1.** Source data for *Figure 8*.
DOI: https://doi.org/10.7554/eLife.35269.029

**Figure supplement 1.** Low concentration of TTX does not affect glutamate release from presynaptic nerve terminals.
DOI: https://doi.org/10.7554/eLife.35269.028

**Figure supplement 1—source data 1.** Source data for *Figure 8—figure supplement 1*.
DOI: https://doi.org/10.7554/eLife.35269.030

STDP (*Figure 2* and *Figure 2—figure supplement 1*), which is consistent with recent reports (*Yang and Dani, 2014*; *Lopez-Rojas et al., 2016*).

However, several studies have demonstrated EPSP-AP pairing protocol-induced synaptic potentiation at the same synapses (*Levy and Steward, 1983*; *Lin et al., 2006*). Given that *Levy and Steward (1983)* and *Lin et al. (2006)* employed in vivo and in vitro field recording configurations, respectively, the discrepant results could be attributed to GCs at different stages of maturation as immature GCs exhibit a lower threshold for LTP induction (*Schmidt-Hieber et al., 2004*; *Ge et al., 2007*) or under different recording circumstances that were exposed to various neuromodulators. For example, *Yang and Dani (2014)* reported that the pairing protocol that showed no synaptic potentiation could induce reliable LTP at PP-GC synapses after D1-type dopamine receptor activation by which dendritic A-type $K^+$ currents are suppressed. Because $I_A$ is known to limit the backpropagation of APs (*Hoffman et al., 1997*), suppression of $I_A$ can boost AP backpropagation, allowing sufficient dendritic depolarization for LTP induction at distal synapses. Our observations of a high density of dendritic $I_A$ and their effects on AP backpropagation in GCs directly support the finding of *Yang and Dani (2014)*. Therefore, under physiological conditions, long-lasting dendritic depolarization during theta oscillation (*Buzsáki, 2002*) or activation of neuromodulatory systems (*Hamilton et al., 2010*; *Yang and Dani, 2014*) may cause inactivation of dendritic $I_A$ and trigger pairing-induced LTP (*Lin et al., 2006*; *Brunner and Szabadics, 2016*).

Although the presence of A-type $K^+$ channels in GC dendrites had been shown in earlier immuno-cytochemical studies (*Birnbaum et al., 2004*; *Monaghan et al., 2008*; *Menegola et al., 2008*), it has been proposed that dendritic A-type $K^+$ channels have no significant impact on bAP-induced $Ca^{2+}$ signals (*Krueppel et al., 2011*). *Krueppel et al. (2011)* results appear to be inconsistent with our present data, which show the strong expression of functional A-type $K^+$ channels in GC dendrites. In contrast to that study, we directly tested the effect of 4-AP in simultaneous soma-dendrite recordings by using both global and local application methods. Our results show that the effect of local puff application of 4-AP to the dendritic recording site is much smaller than that of global application. As *Krueppel et al. (2011)* used local application halfway between the soma and the $Ca^{2+}$ imaging site, the impact of 4-AP on bAPs in their study could be underestimated.

We further found that high-frequency stimulation of PP synapses is efficient in eliciting dendritic spikes required for LTP induction. As reported in a recent in vivo study, mature GCs are exposed to abundant functional glutamatergic inputs from the entorhinal cortex during theta rhythm (*Pernía-Andrade and Jonas, 2014*; *Schmidt-Hieber et al., 2014*). Moreover, several lines of evidence demonstrated that mature GCs are highly innervated by PP synaptic connections while receiving a powerful perisomatic inhibition (*Dieni et al., 2013*; *2016*; *Temprana et al., 2015*). Under these in vivo network conditions, strong dendritic excitation by high-frequency PP inputs and strong perisomatic inhibition may promote amplification of dendritic responses without axosomatic APs. Therefore, dendritic spikes are likely physiologically relevant signals for the induction of LTP. Support for the idea of cooperative LTP at PP-GC synapses has also come from *McNaughton et al. (1978)*, who demonstrated that high-frequency stimulation of PP synapses could induce synaptic enhancement in the absence of GC discharges.

This specific high-frequency stimulation-dependent synaptic potentiation in GCs presumably stems from distinct functional and geometrical features of GC dendrites. A moderate density of dendritic $Na^+$ channels together with higher input impedance of the distal dendrites (*Hama et al., 1989*; *Schmidt-Hieber et al., 2007*; *Holmes, 1989*) suffices for initiating spikes locally in distal dendrites that have small capacitive load but not for supporting active backpropagation of axosomatic APs. Although GCs have relatively short dendrites, a high $K^+$ to $Na^+$ current ratio in these thin-caliber dendrites imposes a strong distance-dependent attenuation of axosomatic APs, leading to a lack of pairing-induced LTP at distal synapses. Thus, $Na^+$ spikes in the dendrites contribute the postsynaptic depolarization necessary for the induction of associative plasticity (*Golding et al., 2002*; *Kim et al., 2015*). However, it should be noted that dendritic $Na^+$ spikes were accompanied by NMDAR-mediated plateau potentials and therefore these two dendritic events could act in concert to trigger LTP in GCs (*Schiller et al., 2000*).

Consequently, our findings suggest that in the absence of axonal firing (*Alme et al., 2010*; *Diamantaki et al., 2016*), dendritic spikes would allow a silent GC to participate in the storage of memories via LTP induction. It would further permit a functional separation between a storage phase

mediated by Na$^+$ channels in GC dendrites, and a recall phase (*O'Neill et al., 2008*) that effectively activates the CA3 neurons (*Vyleta et al., 2016*) via Na$^+$ channels in GC axons.

## Materials and methods

### Slice preparation and electrophysiology

Acute hippocampal slices (thickness, 350 µm) were prepared from the brains of 17- to 25-day-old Sprague-Dawley rats of either sex. Rats were anesthetized (isofluorane, Forane; Abbott) and decapitated rapidly. All the experiments were approved by the University Committee Animal Resource in Seoul National University (Approval #: SNU-090115–7). All brains were sliced coronally, and dorsal slices displaying all the subregions of the hippocampal formation were used for the experiments (Coronal sections located between 4.3 mm and 5.7 mm from the posterior end of the right hemisphere). Slices were prepared in an oxygenated ice-cold sucrose-containing physiological saline using a vibratome (VT1200, Leica), incubated at ~36°C for 30 min, and subsequently maintained in the same solution at room temperature until the recordings. Recordings were performed at near-physiological temperature (33–35°C) in an oxygenated artificial cerebral spinal fluid (ACSF).

Patch pipettes were made from borosilicate glass capillaries (outer diameter = 1.5 mm, inner diameter = 1.05 mm) with a horizontal pipette puller (P-97, Sutter Instruments). The open-tip resistance of patch pipettes was 2.5–6.5 MΩ and 11–30 MΩ for somatic and dendritic recordings, respectively. Current-clamp recordings were performed with an EPC-10 USB Double amplifier (HEKA Elektronik). In current-clamp recordings, series resistance was 8–80 MΩ. Pulse protocols were generated, and signals were low-pass filtered at 3 or 10 kHz (Bessel) and digitized (sampling rate: 20–50 kHz) and stored using Patchmaster software running on a PC under Window 10. Resting membrane potential (RMP) was noted immediately after rupture of the patch membrane. $R_{in}$ was determined by applying Ohm's law to the steady-state voltage difference resulting from a current step (±50 pA). Pipette capacitance and series resistance compensation (bridge balance) were used throughout current-clamp recordings. Bridge balance was checked continuously and corrected as required. Experiments were discarded if the resting membrane potential depolarized above –70 mV and were stopped if the resting membrane potential or $R_{in}$ changed by more than 20% during the recording.

All experiments were performed on visually identified mature GCs on the basis of the relatively large and round-shaped somata, and the location of the cell body under DIC optics. GCs located at the superficial side of the GC layer in the suprapyramidal blade were preferentially targeted. These cells had the average RMP of –81.3 ± 0.2 mV and $R_{in}$ of 108.4 ± 2.6 MΩ (n = 165), that is similar to characteristic intrinsic properties of mature GC population (*Schmidt-Hieber et al., 2004*). Cells were filled with a fluorescent dye, Alexa Fluor 488 (50 µM, Invitrogen) and imaged with an epifluorescence system mounted on an upright microscope equipped with a 60 x (1.1 N.A.) water immersion objective lens. Focal electrical stimulation (100 µs pulses of 1–35 V) was applied on isolated dendrites in the outer third of the molecular layer (within 100 µm of the hippocampal fissure) by placing a glass microelectrode (0.5–3 MΩ) containing 1 M NaCl or ACSF in the vicinity of the selected dendrite (typically at <50 µm distance), guided by the fluorescent image of the dendrite. All experiments were performed in the presence of the GABA receptor antagonist picrotoxin (PTX, 100 µM) and CGP52432 (1 µM).

To record voltage-gated Na$^+$ or K$^+$ currents, outside-out patches were excised from the soma and the dendrite with pipettes of similar geometry and open-tip resistances (18.8 ± 0.4 MΩ, n = 60, ranging from 12.5 to 24.4 MΩ) for comparison of channel density between soma and dendrite (*Kim et al., 2012*). Ensemble K$^+$ currents were evoked by a pulse protocol consisting of a 50–200 ms prepulse to –120 mV followed by a 200 ms test pulse to 70 mV. Na$^+$ currents were generated by a pulse sequence comprised of a 100 ms prepulse to –120 mV and a 30 ms test pulse to 0 mV. In all cases, the holding potential was –90 mV before and after the pulse protocol. Voltage protocols were applied to outside-out patches once every 3 and 5 s for Na$^+$ and K$^+$ current recordings, respectively. Leak and capacitive currents were subtracted online using the pipette capacitance compensation circuit of the amplifier and a 'P over –4' correction procedure.

## Subcellular dendritic patch-clamp recording

Dendritic recordings from GCs were obtained similarly as described previously (*Kim et al., 2012*). First, Alexa Fluor 488 (50 μM, Invitrogen) diluted in an internal solution was loaded into cells via a somatic recording pipette. Second, after ~10 min of loading, fluorescently labeled dendrites were traced from the soma into the molecular layer using epifluorescence microscope. Finally, fluorescent and infrared differential interference contrast (IR-DIC) images were compared, and GC dendrites were patched under IR-DIC.

## Stimulation protocols for the induction of long-term potentiation (LTP)

LTP was induced by either theta burst stimulation (TBS; *Schmidt-Hieber et al., 2004*) or pairing protocols. TBS induction protocol consisted of burst of EPSPs (10 stimuli at 100 Hz) repeated 10 times at 5 Hz. These episodes were repeated four times every 10 s. The pairing protocol comprised 300 repetitions of a presynaptic stimulation and one or two postsynaptic APs at the different time intervals at 1 Hz. A postsynaptic AP was evoked by a brief current injection to the soma (2 ms, 3 nA). For LTP experiments, baseline EPSPs evoked by stimulating presynaptic axon fibers at 0.1 Hz were measured for ~10 min after whole-cell recording. After LTP induction protocol, EPSPs were recorded for 30 min. LTP magnitude was evaluated as the percentage of EPSP baseline (5 min) after 25 to 30 min after the induction protocol. In a subset of recordings, local application of TTX (1 μM) to the perisomatic area was used to prevent axosomatic AP initiation and backpropagation during TBS induction, without affecting evoked EPSPs.

## Solutions and chemicals

The extracellular solution for dissection and storage of brain slices was sucrose-based solution (87 mM NaCl, 25 mM NaHCO$_3$, 2.5 mM KCl, 1.25 mM NaH$_2$PO$_4$, 7 mM MgCl$_2$, 0.5 mM CaCl$_2$, 10 or 25 mM glucose, and 75 sucrose). Physiological saline for experiments was standard ACSF (125 mM NaCl, 25 mM NaHCO$_3$, 2.5 mM KCl, 1.25 mM NaH$_2$PO$_4$, 1 mM MgCl$_2$, 2 mM CaCl$_2$, and 25 mM glucose). TTX and 4-AP were applied either via bath perfusion (0.5 μM and 10 mM, respectively) or by local application (1 μM and 10 mM, respectively) with a pressure application system (Picospritzer 3, General Valve). Pressure pulses had durations of 0.2 s and amplitudes of ~10 psi. CdCl$_2$ and NiCl$_2$ were applied in the bath at a concentration of 200 μM and 50 μM, respectively.

For whole-cell recording and K$^+$ current recording in outside out patches, we used K$^+$ rich intracellular solution that contained 115 mM K-gluconate, 20 mM KCl, 10 mM HEPES, 0.1 mM EGTA, 2 or 4 mM MgATP, 10 mM Na$_2$-phosphocreatine, and 0.3 mM NaGTP, pH adjusted to 7.2–3 with KOH (~300 mOsm). In a subset of experiments, 50 μM Alexa 488 and 0.1–0.2% biocytin (wt/vol) were added to the internal solution for labeling during the experiment or after fixation, respectively. All drugs were dissolved in physiological saline immediately before the experiment and perfused on slices at a rate 4–5 ml min$^{-1}$. These included: DL-AP5, PTX, CGP52432, TTX, 4-AP, and TEA. Intracellularly delivered QX-314 (N-(2,6-dimethylphenylcarbamoylmethyl) triethylammonium chloride) were directly added to the pipette solution before the experiment was started. TTX and DL-AP5 (D, L-2-amino-5-phoshonovaleric acid; 50 μM) were purchased from Tocris Bioscience; CGP52432 were from Abcam; All other drugs were from Sigma-Aldrich.

## Immunohistochemistry

Post-hoc morphological analysis of GCs was performed as described previously with slight modifications (*Kim et al., 2012*). Slices were fixed overnight at 4°C in 4% paraformaldehyde in 100 mM phosphate buffer solution (PBS), pH7.4. After fixation, slices were rinsed several times with PBS and permeabilized with 0.3% Triton X-100 in PBS. Subsequently, slices were treated with 0.3% Triton X-100% and 0.5% BSA in PBS to prevent any unwanted. Next, slices were treated with 0.3% Triton X-100% and 0.2% streptavidin-cy3 in PBS and were again incubated overnight in 4°C. Finally, slices were mounted with DAKO S3023 medium.

## Data analysis

Custom-made routines written in Igor Pro 6.3 (Wavemetrics), Stimfit 0.15 (*Guzman et al., 2014*) or Prism (Graphpad) were used for data analysis and statistical testing. Spike threshold value was determined as the time point at which the derivative of voltage exceeded 40 V/s at the soma or 20 V/s at

the dendrite. To determine the conduction velocity, latency–distance data were fit with a linear regression, and the velocity was calculated from the linear regression slope.

All values indicate mean ± standard error of the mean (SEM), with '*n*' denoting the number of experiments. To test statistical significance, a two-sided nonparametric Wilcoxon signed-rank test or Wilcoxon rank sum test were used (unless noted otherwise). For comparisons of more than two groups, a Kruskal-Wallis test with Dunn's multiple comparisons for post-hoc testing was used. Differences with p-value less than 0.05 were indicated with an asterisk and considered significant. Membrane potential values were displayed without correction for liquid junction potentials. Distances were measured from the soma to the dendritic recording site along the dendritic trajectory (*Kim, 2014*).

## Acknowledgements

We thank Jose Guzman and Hua Hu for critically reading the manuscript.

## Additional information

### Funding

| Funder | Grant reference number | Author |
| --- | --- | --- |
| National Research Foundation of Korea | NRF-2015R1C1A1A02037776 | Sooyun Kim |
| Ministry of Education | Brain Korea 21 PLUS Program | Sooyun Kim |
| National Research Foundation of Korea | NRF-2010-0027941 | Won-Kyung Ho |
| National Research Foundation of Korea | NRF-2017R1A2B2010186 | Won-Kyung Ho |

The funders had no role in study design, data collection and interpretation, or the decision to submit the work for publication.

### Author contributions

Sooyun Kim, Conceptualization, Resources, Data curation, Formal analysis, Supervision, Funding acquisition, Validation, Investigation, Visualization, Methodology, Writing—original draft, Project administration, Writing—review and editing; Yoonsub Kim, Formal analysis, Investigation, Writing—review and editing; Suk-Ho Lee, Resources, Writing—review and editing; Won-Kyung Ho, Resources, Supervision, Funding acquisition, Writing—review and editing

### Author ORCIDs

Sooyun Kim http://orcid.org/0000-0002-2035-3247
Won-Kyung Ho http://orcid.org/0000-0003-1568-1710

### Ethics

Animal experimentation: This study was performed in strict accordance with the recommendations in the Guide for the Care and Use of Laboratory Animals of the Seoul National University. All of the animals were handled according to approved institutional animal care and use committee (IACUC) of the Seoul National University. The protocol (Approval #: SNU-090115-7) was approved by the Committee on the Ethics of Animal Experiments of the Seoul National University. Animals were anesthetized by inhalation of 5% isoflurane before sacrifice, and every effort was made to minimize suffering.

### Decision letter and Author response

Decision letter https://doi.org/10.7554/eLife.35269.039
Author response https://doi.org/10.7554/eLife.35269.040

## Additional files

**Supplementary files**
• Transparent reporting form
DOI: https://doi.org/10.7554/eLife.35269.031

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
