## [Decision Letter]

Thank you for submitting your article "Dendritic spikes in hippocampal granule cells are necessary for long-term potentiation at the perforant path synapse" for consideration by *eLife*. Your article has been reviewed by three peer reviewers, and the evaluation has been overseen by a Reviewing Editor and Gary Westbrook as the Senior Editor. The following individual involved in review of your submission has agreed to reveal his identity: Maarten Kole (Reviewer #1). The reviewers have discussed the reviews with one another and the Reviewing Editor has drafted this decision to help you prepare a revised submission.

Summary:

This report documents Nav-dependent electrogenesis in dendrites of dentate gyrus granule cells and shows how this electrogenesis is critical for perforant path long-term potentiation. The experiments are technically demanding and the overall work is of high quality, and should be published. However, the reviewers have several concerns that need to be addressed. Notably, there was a general concern that the paper is over-hyped, and there is really not a need for this. The results are clean and should be simply presented to document the findings.

Essential revisions:

1) There is a lack of clarification and experimentation why the observed "high" Ia-type current density blocks excitation in the axial direction but still enables local dendritic Na^+^ spikes. The ion channel distribution profiles alone (Figure 5) are insufficient to explain this since direct experiments addressing their role in spike-timing dependent failures is lacking. It is neither discussed nor addressed experimentally how the specific granule cell cable structure contributes. The interplay between ionic currents and the passive cable properties are the true backbone upon which depolarizations either spreads or attenuates. One prediction is that local block of IA (e.g. with 4-AP) would cause back-propagation facilitation without changing local dendritic spikes. Dual whole-cell voltage recording during synaptic stimulation and local somatic 4-AP application would be one way to address this issue.

2) The pharmacology experiments in Figure 6—figure supplement 1 are flawed (subsection “Dendritically initiated Na^+^ spikes are required for TBS-induced LTP at PP–GC synapses”, first paragraph). Dendritic spikes are expected to be mediated by nickel-sensitive T-type Ca^2+^ channels (at 50 – 100 µM, see e.g. Schmidt-Hieber, Jonas and Bischofberger,et al. 2004) and insensitive to cadmium (a high voltage-activated calcium channel blocker). Experiments with nickel (or TTA-P2) would more firmly demonstrate what the precise pharmacology of the observed dendritic spike is; sodium, T-type calcium channels or both. In fact, it seems cadmium did block a late component in the depolarization in the dendritic recording? Please quantify and clarify.

3) The role of dendritic spikes is introduced with a lot of confidence in the Abstract as a 'key mechanism for memory formation'. However, it remains unexplained why the authors see it in some but not all DG cells. The age range of 17-24 days is not 'mature' (subsection “Slice preparation and electrophysiology”, third paragraph). Only in the aforementioned subsection it is mentioned that morphology may be different for the DG cells showing dendritic spikes but quantification and details are lacking. Were dendritic spikes seen in cells with different input resistances? Were there any electrophysiological or morphological correlates? Given the well-known neurogenesis in this region the authors will need to identify the DG cells and show whether dendritic spikes are a true general phenomenon or limited to young immature cells.

4) The authors find that the expression of LTP is correlated with the presence of fast rising events at the soma during induction, which they suggest reflects the generation of dendritic spikes. These somatic events need to be better quantified (see also point 5). What were their kinetics and amplitude? How many of these events were "necessary" for LTP to be observed? Was there a correlation between the number of these events and the magnitude of LTP?

5) The experiment showing that blocking somatic APs using local somatic TTX applications had no effect on LTP induction is a critical. Yet, it is not well documented. It is only mentioned briefly in the subsection “LTP by TBS at PP–GC synapses requires NMDARs and Na^+^ channels”, with data apparently pooled with other data in Figure 1D.

6) The rationale for Figure 4 is unclear. Firstly, the impact of the time taken for APs to propagate from the soma to the distal dendrites (1 ms) is minimal. Secondly, why were timings of 1 to 3 ms used in these experiments? Classical STDP is induced using +/- 10 ms timing as the +10 ms time coincides (approximately) with when an AP would be evoked by a synaptic input. What is the rational for +1 and +3 ms timing? In its current form this figure is of limited value.

7) Figure 6 shows, using dual somatic and dendritic recording, that isolated dendritic Na spikes can be evoked at distal dendritic locations during dendritic current injection and importantly also during TBS stimulation. These data potentially provide evidence that the fast rising events observed at the soma during LTP induction are indeed dendritic spikes. If true, at the soma these events should have the same properties and characteristics. Was this the case? This analysis is critical to the argument that dendritic spikes are necessary for LTP induction.

8) In regard to the data in Figure 6, while not essential, it would have been good to show using dendritic recording that APV blocks dendritic spikes evoked by TBS stimulation. This would explain why the somatic representation of these events is absence in APV and why LTP was blocked under these conditions.

9) The argument made in the third paragraph of the Discussion that dendritic K channels may limit AP backpropagation is inconsistent with the observations of Krueppel et al. (2011), who found that 5 mM 4-AP had no impact on dendritic calcium transients mediated by bAPs.

10) There is an overly hyped description of the results, and issues with some of the interpretations of the data:

A) LTP is *not* learning, synaptic or behavioral. LTP is a form of long-term plasticity that certainly plays a role in learning. The authors go way overboard in selling the significance of their results. Some examples of this are:

Abstract: "associative learning at distal synapses"

Introduction, first paragraph: "synaptic learning" "similar memories"

Discussion, first paragraph: "formation of memories"

Discussion, fourth paragraph: "synaptic learning"

B) I suggest that the authors rewrite these sections more in keeping with the role of LTP as one (of many) cellular/synaptic changes that are important for learning and memory.

11) I would quibble with the authors' conclusions that dendritic spikes are necessary and/or required for LTP. For example, if Na channels were blocked, could LTP be induced if there was still sufficient depolarization in the dendrites? I suspect that the answer to this is yes. Furthermore, in each of the examples shown of a dendritic spike (Figure 1D, Figure 1—figure supplement 1, Figure 6D) there appears to be an accompanying plateau potential, presumably due to NMDA receptors. So, which is required, the Na spike or plateau potential, or both. Given data from other cell types, perhaps the Na spike facilitates the triggering of an NMDA plateau, both of which are required or necessary for LTP induction "under physiological conditions".

12) In Figure 3 the authors explore spike back propagation from soma into dendrites. A plot of amplitude vs distance is shown in Figure 3E. These results are similar to those from Krueppel et al. However, I think the authors should also plot the amplitude as a function of relative distance as Krueppel et al. did in their Figure 1E. This is useful for comparing bAP spike amplitude with other cell types that have different dendritic lengths. It looks to me that the attenuation is not that different (as a function of total length) from pyramidal cells, so I would rewrite the sentence “The bAP attenuation was more pronounced (length constant of 182 µm; n = 46) than in the dendrite of other hippocampal principal neurons (Spruston et al., 1995; Golding et al., 2002; Kim et al., 2012)”. I would also suggest an experiment in which they measure bAP amplitude before and after adding 4AP in the bath to test whether the bAP amplitude is sensitive to block of A-type K channels. Their data in Figure 5 suggest that it would be but Krueppel suggest not. Otherwise, if Na channels have uniform density (their Figure 5), what is the mechanism for the declining bAP amplitude with distance.

13) The data from outside-out patches and shown in Figure 5 are very nice (and impressive) given the small size of the dendrites. However, the figure legend states that all the recordings were performed with 5 mM 4AP and 20 mM TEA in the bath. If this is correct, what is the outward current shown in 5A?

14) While the authors suggest a "unique" distribution of Na and K channels (Abstract), one could argue that their results for GCs are qualitatively similar to those from CA1 pyramidal cells, but different from the conclusions of Krueppel et al. The results presented are nonetheless important since so little is known about DGC dendrites.

[Editors’ note: this article was subsequently rejected after discussions between the reviewers, but the authors were invited to resubmit for further consideration.]

Thank you for submitting your work entitled "Dendritic spikes in hippocampal granule cells are necessary for long-term potentiation at the perforant path synapse" for consideration by *eLife*. Your article has been reviewed by three peer reviewers, and the evaluation has been overseen by a Reviewing Editor and a Senior Editor. The following individuals involved in review of your submission have agreed to reveal their identity: Dan Johnston (Reviewer #3).

Our decision has been reached after consultation between the reviewers. Based on these discussions and the individual reviews below, we regret to inform you that your work will not be considered further for publication in *eLife*. Although there is significant novelty in the new findings regarding potential roles of dentate gyrus granule cell dendritic spikes in plasticity, numerous issues remain in this revision. Thus at this point we cannot accept this manuscript for publication, however we would be willing to consider a new submission if you decide to fully address the remaining critiques.

*Reviewer #1:*

In this revision, Kim and co-authors have addressed most of my major concerns and provided new experimental data in support of the general conclusion that distal dendritic sodium-mediated spikes contribute to LTP at the perforant path (PP) synapses. The manuscript has improved and in my opinion builds a strong case for dendritic spike generation. Although the experiments are in general compelling it still contains errors and statements sometimes lack quantification (see below).

Regarding the request for more detail how the authors calculated outside-out surface area the authors write, "we used Hu and Jonas, 2014". But when reviewing Figure 5 and subsection “Ionic mechanisms of AP backpropagation”, there is an obvious mistake, which was present already in the first version of this manuscript. The y-axis shows negative values for sodium conductance density, which is biophysically implausible. I have the impression that the authors rather plotted current density (unit amperes per square micrometre) but labelled their axes with conductance density (unit siemens). This error seems to propagate through the data presentation for the A-type and delayer rectifier K^+^ density distributions. I don't think that a correction for driving force was applied based on the numbers and it is neither clear whether and how area was corrected for. The authors may want to consult the article of Schmidt-Hieber and Bischofberger, 2010 (J. Neurosci. 30(30); p. 10233), read also the supplement and thoroughly revisit this part of their study. As requested earlier, it is important they spell out how data were obtained and compare their numbers with previously published data. Both for the sodium and potassium measurements.

*Reviewer #2:*

Overall the authors have done a good job of addressing the points raised by the reviewers. Nevertheless, I have suggestions for additional changes to the manuscript that do not require new experiments:

1) I was a little surprised to see no clear correlation between the number of putative dendritic spikes observed at the soma during TBS stimulation and the magnitude of LTP (Author response image 3). Also, the number of putative dendritic spikes associated with LTP induction seems very low. As the authors indicate this may be because they have missed detecting dendritic spikes in their somatic recordings, as suggested by the data shown in Author response image 4. The capacity to detect dendritic spikes in their somatic recordings is key to the idea that LTP induction requires dendritic spike generation, as discussed in the subsection “TBS-induced LTP at the PP-GC synapses does not require postsynaptic bAPs” and concluded from the data in Figure 1. As indicated in my original review, a characterisation of the somatic events detected when dendritic spikes are observed directly during dendritic recordings would provide direct evidence that the fast rising events observed at the soma during TBS stimulation are indeed dendritic spikes. It would also be important in my opinion to quantify how reliably dendritic spikes can be detected by somatic recordings.

2) Some of the supplemental data is key to the story in my opinion. In particular, I am referring to the data in Figure 1—figure supplements 1 and 2. I suggest parts (or all) of Figure 1—figure supplement 1 is included in Figure 1, and parts (or all) of Figure 1—figure supplement 2 is included in Figure 2.

3) I still think the rational for Figure 4 is weak. This figure does not add much to the paper. The observed bAP conduction velocities are not that different from that seen in pyramidal cells where standard +10/-10 ms EPSP-AP timing protocols can evoked STDP at proximal synapses.

*Reviewer #3:*

The authors have nicely addressed my previous concerns, and I have no other major comments.

[Editors’ note: what now follows is the decision letter after the authors submitted for further consideration.]

Thank you for resubmitting your work entitled "Dendritic spikes in hippocampal granule cells are necessary for long-term potentiation at the perforant path synapse" for further consideration at *eLife*. Your revised article has been favorably evaluated by Gary Westbrook (Senior Editor), a Reviewing Editor, and two reviewers. Both reviewers think that the manuscript has been improved, yet additional clarification is needed, as support for the main conclusions of the paper requires additional analysis and improved presentation.

Specific points:

1) There is a little confusion in the first section of the Results ("TBS-induced LTP at the PP-GC synapses does not require postsynaptic bAPS"). The authors first state "To ensure that no axosomatic AP initiation and backpropagation occur during TBS, we locally applied tetrodotoxin (TTX) to the GC axon, soma, and proximal dendrites in a subset of experiments (6 out of 13 experiments)", then later you say "To test the contribution of bAPs in this form of LTP, we applied strong synaptic stimulation without perisomatic TTX application". It would be better to first discuss and show the control case in the absence of TTX, describe as you do the presence of putative dendritic spikes during synaptic stimulation and how these are correlated with the magnitude of LTP, and only then introduce the idea that bAPs are not required showing both that the magnitude of LTP is not related to the presence of APs under control conditions and that LTP persists when APs were blocked.

2) The finding that APV "abolished LTP”, but had little if any impact on putative dendritic spikes (Figure 2B and subsection “LTP by TBS at PP–GC synapses requires NMDARs and Na^+^ channel”: "fast rising events remained unchanged") questions the causal role of these putative dendritic spikes in LTP induction. The authors address this by using QX-314 to block Na^+^ channels, finding that this "abolished both plateau potentials and TBS-induced LTP". Do the authors mean "abolished both putative dendritic spikes and TBS-induced LTP"? Assuming this is the case some quantification of the effect of QX-314 on putative dendritic spikes is warranted (E.g. data on the number of putative dendritic spikes in control versus QX-314). Note, also the figure reference in the aforementioned subsection should be Figure 2C, D not Figure 2B-D.

3) The authors state that "Pooled data demonstrated that a moderate density of Na^+^ channels is distributed over the dendritic membrane", yet then indicate dendritic Na^+^ current densities of between 136 pS/um^2 (proximal) and 206 pS/um^2 (distal). These are *high* not moderate densities. One model of AP backpropagation in granule cells Krueppel et al. (2011) used dendritic Na^+^ channel densities of around 2 mS/cm^2 (or 20 pS/um^2). That is, almost a factor of 10 lower than estimated by the authors. The estimated densities of A-type and delayed rectifier type K^+^ channels are also very high. Given some uncertainty in the accuracy of estimating the patch membrane area based on pipette capacitance, it might therefore be better to simply state the peak current amplitude rather than current density.

4) Addition of the data showing the association between putative dendritic spikes observed at the soma (based on dV/dt) and the direct observation of dendritic spikes during dendritic recordings is most welcome. In my view this data (Figure 5—figure supplement 3) is critical to the paper and therefore should not be "buried" in the supplemental data, but should be included as part of Figure 5 of the manuscript. I would argue the data in Figure 1—figure supplement 1, showing that APs are not required for LTP, is also critical to the story and therefore should also be presented as a main figure rather than supplemental data.

[Editors' note: further revisions were requested prior to acceptance, as described below.]

Thank you for resubmitting your work entitled "Dendritic spikes in hippocampal granule cells are necessary for long-term potentiation at the perforant path synapse" for further consideration at *eLife*. Your revised article has been favorably evaluated by Gary Westbrook (Senior Editor), and the Reviewing Editor, John Huguenard.

The manuscript has been improved but there are a few remaining issues that need to be addressed before acceptance, as outlined below:

1) There is some confusion regarding the figure labeling in the Results text, especially around new Figure 5 (formerly Figure 4, subsection “Ionic mechanisms of AP backpropagation”). Please check carefully to ensure that figures are cited correctly.

2) Although strong dendritic spikes are defined as > 10 mV/ms in the subsection “TBS-induced LTP at the PP-GC synapses does not require postsynaptic bAPs”, the definition of weak dendritic spikes is more ambiguous, especially given that some thresholds must have been set to distinguish between EPSPs and weak dendritic spikes. This discriminant belongs in the text. The text table in Figure 7B suggests it is on the order of 3 mV/ms as the minimum somatic detection of a DS.

---

## [Author Response]

Essential revisions:1) There is a lack of clarification and experimentation why the observed "high" Ia-type current density blocks excitation in the axial direction but still enables local dendritic Na^+^ spikes. The ion channel distribution profiles alone (Figure 5) are insufficient to explain this since direct experiments addressing their role in spike-timing dependent failures is lacking. It is neither discussed nor addressed experimentally how the specific granule cell cable structure contributes. The interplay between ionic currents and the passive cable properties are the true backbone upon which depolarizations either spreads or attenuates. One prediction is that local block of IA (e.g. with 4-AP) would cause back-propagation facilitation without changing local dendritic spikes. Dual whole-cell voltage recording during synaptic stimulation and local somatic 4-AP application would be one way to address this issue.

We thank the reviewer for this suggestion. Bath application of 4-AP increased both the peak amplitude (142 ± 2.6%, n = 6) and the duration (378 ± 59%, n = 6) of backpropagating APs in the dendrites. It indicates that A-type K^+^ channels affect AP backpropagation, as the reviewer suggested (now presented in Figure 5—figure supplement 1 in the revised manuscript). Next, focal 4-AP application at the dendritic site lowered the current threshold for dendritic spike initiation (Control, 0.84 ± 0.15 nA; 4-AP, 0.76 ± 0.14 nA, n = 8, *P* < 0.005), suggesting that A-type K^+^ channels may contribute to the regenerative dendritic spikes by, for example, confine them to local dendritic locations (now in Figure 6—figure supplement 1 in the revised manuscript). We now discuss these findings in the Results subsections “Ionic mechanisms of AP backpropagation” and “Dendritically initiated Na^+^ spikes are required for TBS-induced LTP at PP–GC synapses”.

Finally, we also discuss the effect of the GC dendritic morphology on the specific dendritic function as suggested (Discussion, sixth paragraph).

2) The pharmacology experiments in Figure 6—figure supplement 1 are flawed (subsection “Dendritically initiated Na^+^ spikes are required for TBS-induced LTP at PP–GC synapses”, first paragraph). Dendritic spikes are expected to be mediated by nickel-sensitive T-type Ca^2+^ channels (at 50 – 100 µM, see e.g. Schmidt-Hieber, Jonas and Bischofberger, 2004) and insensitive to cadmium (a high voltage-activated calcium channel blocker). Experiments with nickel (or TTA-P2) would more firmly demonstrate what the precise pharmacology of the observed dendritic spike is; sodium, T-type calcium channels or both. In fact, it seems cadmium did block a late component in the depolarization in the dendritic recording? Please quantify and clarify.

As the reviewer cited, T-type calcium spikes are observed in newly generated granule cells (i.e., cells with input resistance > 1.5 GΩ, Schmidt-Hieber et al., 2004, Figure 3). Our dataset includes only mature granule cells (input resistance: 108.0 ± 2.6 MΩ). As expected, bath application of 50 µM NiCl_2_ did not block dendritic spikes (97.9 ± 10.2% of peak amplitude, 4 experiments).

Bath application of 200 µM CdCl_2_ did not affect the magnitude of the late-depolarizing response of the dendritic spikes evoked by brief current injection (98.9 ± 14.7% of baseline, 3 experiments), and documented it in the legend of Figure 6—figure supplement 1in the revised manuscript. We think that this response is due to the morphological and passive electrical properties of the GC dendrites (Author response image 1).

We acknowledge the suggestion of the reviewer and add now the nickel experiments in Figure 6—figure supplement 1and the cadmium effects in the legend. In this way, we hope to emphasize the sodium origin of the dendritic spikes.

**Author response image 1. respfig1:** Effects of Cd^2+^ on a late component in the dendritic depolarization. (A, B) Somatic (black) and dendritic (red) voltage responses to dendritic current injection pulses with increasing amplitude (A: 200 pA; B; 700 pA). Voltage responses in the presence of CdCl_2_ were recorded 10 minutes after bath application of CdCl_2_. Note the decrease in the subthreshold depolarizing response of the dendritic trace after bath application of CdCl_2_ in A. (C) Summary of the effect of CdCl_2_ on the late depolarizing component in 3 dendritic recordings, indicating that CdCl_2_ did not affect the magnitude of the late depolarizing phase of the dendritic spikes.

3) The role of dendritic spikes is introduced with a lot of confidence in the Abstract as a 'key mechanism for memory formation'. However, it remains unexplained why the authors see it in some but not all DG cells. The age range of 17-24 days is not 'mature' (subsection “Slice preparation and electrophysiology”, third paragraph). Only in the aforementioned subsection it is mentioned that morphology may be different for the DG cells showing dendritic spikes but quantification and details are lacking. Were dendritic spikes seen in cells with different input resistances? Were there any electrophysiological or morphological correlates? Given the well-known neurogenesis in this region the authors will need to identify the DG cells and show whether dendritic spikes are a true general phenomenon or limited to young immature cells.

We thank the reviewer for pointing out these issues. To address the reviewers points, we have recorded newly generated GCs (>1GΩ, see Schmidt-Hieber et al., 2004, Figure 1E). Unlike mature GCs (<200 MΩ), dendritic spikes were not observed in response to high-frequency synaptic stimulation of PP synapses (Figure 2 for reviewers; 4 cells). Direct dendritic recordings in newly generated GCs is not possible due to its rudimentary dendritic arborization (Schmidt-Hieber et al., 2004, Figure 1A-D). Therefore, we have softened our statements on the developmental effects of the dendritic spike generation (Discussion, first paragraph) and add a more carefully wording on memory formation all over the text. To state clearly that we recorded from mature GC neurons, we detailed the intrinsic electrical properties (such as resting membrane potential and input resistances) of the cells in our dataset in the Results section and revised accordingly the Materials and methods section.

**Author response image 2. respfig2:** High-frequency PP stimulation does not elicit dendritic spikes in young GCs (>1GΩ). (**A**) DIC-image of young GCs located ate the deep side of the GC layer. Scale bar is 50 µm. (**B**) Somatic voltage responses to current pulses (–10 and +10 pA pulses). (**C**) Somatic voltage responses to high-frequency synaptic stimulation of the PP inputs (‘Materials and methods’).

4) The authors find that the expression of LTP is correlated with the presence of fast rising events at the soma during induction, which they suggest reflects the generation of dendritic spikes. These somatic events need to be better quantified (see also point 5). What were their kinetics and amplitude? How many of these events were "necessary" for LTP to be observed? Was there a correlation between the number of these events and the magnitude of LTP?

We provide a definition of the somatic readout of a dendritic spike as the first temporal derivative of the voltage (i.e., dV/dt) which is > 5 mV/ms which is based on a rigorous quantification of our dataset (Figure 1—figure supplement 1C and D of the revised paper). We also analyze the correlation between the number of dendritic spikes and LTP magnitude (Author response image 3) and found no correlation (r = 0.096, *P* = 0.83). We attribute this lack of correlation to the difficulty to unequivocally count dendritic spikes in the somatic recording (Author response image 4).

We add now a sentence in the text, “By examining the temporal derivative (*dV/dt*) of somatic voltage responses, we define dendritic spikes as those with *dV/dt* > 5 mV/ms in all further analysis and experiments (Figure 1—figure supplement 1).”

**Author response image 3. respfig3:** Magnitude of LTP is not correlated with the number of fast rising events at the soma during TBS. Data that were used in this analysis were taken from Figure 1G.

**Author response image 4. respfig4:** Dendritic spikes often appeared completely attenuated at the soma. Dual soma-dendrite recordings reveal that high-frequency synaptic stimulation of the PP inputs sometimes evoked dendritic spikes without a clear corresponding somatic responses, ‘spikelets’.

5) The experiment showing that blocking somatic APs using local somatic TTX applications had no effect on LTP induction is a critical. Yet, it is not well documented. It is only mentioned briefly in the subsection “LTP by TBS at PP–GC synapses requires NMDARs and Na^+^ 140 channels”, with data apparently pooled with other data in Figure 1D.

We apologize for the confusion in the description of the experimental procedures. In Figure 1B-D, somatic APs were absent in all experiments, however perisomatic TTX application was used in a subset of recordings. We have revised the text to describe the experiments clearly (subsection “TBS-induced LTP at the PP-GC synapses does not require postsynaptic bAPs”).

6) The rationale for Figure 4 is unclear. Firstly, the impact of the time taken for APs to propagate from the soma to the distal dendrites (1 ms) is minimal. Secondly, why were timings of 1 to 3 ms used in these experiments? Classical STDP is induced using +/- 10 ms timing as the +10 ms time coincides (approximately) with when an AP would be evoked by a synaptic input. What is the rational for +1 and +3 ms timing? In its current form this figure is of limited value.

We agree with the reviewer that the findings deserve an appropriate explanation. The conduction velocity on the AP in the dendrites is 222 µm/ms (Figure 3). Thus, the AP originated at the soma will arrive at the distal PP synapses (~200–~278 µm from the soma) about one millisecond later. For that reason, we argued that evoking a somatic AP 10 ms after the synaptic stimulation (i.e. classical STDP protocol, Markram et al., 1997) will not provoke a coincidental AP-mediated depolarization with the presynaptic input in the distal GC dendrites (i.e. AP will arrive too late at the synaptic contact). Furthermore, distal dendritic EPSPs in GCs are remarkably brief (~8.8 ms; Schmidt-Hieber et al., 2007). We accounted for both effects by adjusting the induction protocol with short pairings (1–3 ms) expecting to match the synaptic depolarization at the distal GC synapses. We found that it does not produce LTP, and concluded that the somatically initiated AP is not relevant for plasticity at the distal synapses in GCs. We now document this rationale in the Results section (subsection “Backpropagation of axosomatic APs in the dendrites of GCs”) and in the revised Figure 4.

7) Figure 6 shows, using dual somatic and dendritic recording, that isolated dendritic Na spikes can be evoked at distal dendritic locations during dendritic current injection and importantly also during TBS stimulation. These data potentially provide evidence that the fast rising events observed at the soma during LTP induction are indeed dendritic spikes. If true, at the soma these events should have the same properties and characteristics. Was this the case? This analysis is critical to the argument that dendritic spikes are necessary for LTP induction.

We thank the reviewer for this comment. We have characterized the somatic responses when dendritic spikes were elicited. Thus, dendritic spikes can be identified as the first temporal derivative of the somatic voltage which is larger than 5 mV/ms. (now presented in the subsection “TBS-induced LTP at the PP-GC synapses does not require postsynaptic 119 bAPs” and Figure 6B of the revised paper). Please see also our answer #4.

8) In regard to the data in Figure 6, while not essential, it would have been good to show using dendritic recording that APV blocks dendritic spikes evoked by TBS stimulation. This would explain why the somatic representation of these events is absence in APV and why LTP was blocked under these conditions.

We have followed the reviewer’s suggestion and shown that bath-application of 50 µM DL-AP5 reduces plateau potentials at the soma, indicating that they have a NMDAR-component (68.6 ± 5.5% of baseline, 6 experiments; now in Figure 1—figure supplement 2 of the revised paper). Plateau potentials were more prominent at the dendritic locations, as shown in Author response image 5 (2 cells).

**Author response image 5. respfig5:** NMDAR-dependent sustained plateau potentials in dendrites. (**A**) Diagram illustrating the recording configuration of a simultaneous somatic (black amplifier) and dendritic (red amplifier) patch-clamp recording on a GC combined with bath application of the NMDAR blocker, DL-AP5. Scale bar is 50 µm. (**B**) Blockade of NMDARs by a bath application of DL-AP5 (50 μM) abolished sustained plateau potentials. The decrease in plateau potentials were more prominent in the dendrites.

9) The argument made in the third paragraph of the Discussion that dendritic K channels may limit AP backpropagation is inconsistent with the observations of Krueppel et al. (2011), who found that 5 mM 4-AP had no impact on dendritic calcium transients mediated by bAPs.

We have completely revised the role of K^+^ channels in the AP backpropagation and generation of dendritic spikes to the view of our new experiments. It is now documented in Figure 5—figure supplement 1 andFigure 6—figure supplement 2,and discussed in Results subsections “Ionic mechanisms of AP backpropagation” and “Dendritically initiated Na^+^ spikes are required for TBS-induced LTP at PP–GC synapses”, see also our answer #1 for details). To address the concern of the reviewer, we have performed additional experiments using both global and local 4-AP application methods. We found that the effect of focal dendritic application of 4AP on the AP backpropagation (Author response image 6) is virtually smaller than that of global application (see also Figure 5—figure supplement 1). We justify the effects of 4-AP on dendritic spikes due to the focal application procedure of our experiment (Discussion section, fourth paragraph).

**Author response image 6. respfig6:** Effect of dendritic A-type K^+^ channel blockade on AP backpropagation. (**A**) Diagram of the experimental setup illustrating the recording configuration of a simultaneous somatic (black amplifier) and dendritic (red amplifier) patch-clamp recording on a GC combined with focal application of 4-AP directly to the dendritic patch. Scale bar is 50 µm. (**B**) (Left) Sample traces of somatic (black) and dendritic (red) APs evoked by somatic current injection under control (top) and after puff application of 10 mM 4-AP (bottom). (Right) First AP in the train displayed at expanded time scale. Local application of 4-AP to the dendrites selectively affect the duration of APs in the dendrites. (**C**) Summary of the effects of local dendritic application of 4-AP on peak amplitude and duration of APs. (Left; Soma: 100.8 ± 1.3%; Dendrite: 100.6 ± 6.5% ; n = 6, *P* = 0.98; Right; Soma: 105.6 ± 2.3%; Dendrite: 130.5 ± 5.2%; n = 6, ***P < 0.005) in 6 somatodendritic recordings. Bars indicate mean ± SEM; circles represent data from individual cells.

10) There is an overly hyped description of the results, and issues with some of the interpretations of the data:A) LTP is not learning, synaptic or behavioral. LTP is a form of long-term plasticity that certainly plays a role in learning. The authors go way overboard in selling the significance of their results. Some examples of this are:Abstract: "associative learning at distal synapses"Introduction, first paragraph: "synaptic learning" "similar memories"Discussion, first paragraph: "formation of memories"Discussion, fourth paragraph: "synaptic learning"B) I suggest that the authors rewrite these sections more in keeping with the role of LTP as one (of many) cellular/synaptic changes that are important for learning and memory.

Precise wording was given through the text to denote that long-term potentiation is the cellular correlate of learning and memory.

11) I would quibble with the authors' conclusions that dendritic spikes are necessary and/or required for LTP. For example, if Na channels were blocked, could LTP be induced if there was still sufficient depolarization in the dendrites? I suspect that the answer to this is yes. Furthermore, in each of the examples shown of a dendritic spike (Figure 1D, Figure 1—figure supplement 1, Figure 6D) there appears to be an accompanying plateau potential, presumably due to NMDA receptors. So, which is required, the Na spike or plateau potential, or both. Given data from other cell types, perhaps the Na spike facilitates the triggering of an NMDA plateau, both of which are required or necessary for LTP induction "under physiological conditions".

As the reviewer suggested, the dendritic Na^+^ spike may contribute to the NMDAR activation necessary for the induction of LTP (now stated in the Discussion, sixth paragraph). To demonstrate this effect, we dialyzed GC cells with 5 mM QX-314 while stimulating PP synapses at high-frequencies to show that without proper sodium channel activation, it is not possible to induce LTP (92.0 ± 8.1% of EPSP baseline, 8 experiments), as in Mishra et al., 2016. We added a summary graph in Figure 2D.

12) In Figure 3 the authors explore spike back propagation from soma into dendrites. A plot of amplitude vs distance is shown in Figure 3E. These results are similar to those from Krueppel et al. However, I think the authors should also plot the amplitude as a function of relative distance as Krueppel et al. did in their Figure 1E. This is useful for comparing bAP spike amplitude with other cell types that have different dendritic lengths. It looks to me that the attenuation is not that different (as a function of total length) from pyramidal cells, so I would rewrite the sentence “The bAP attenuation was more pronounced (length constant of 182 µm; n = 46) than in the dendrite of other hippocampal principal neurons (Spruston et al., 1995; Golding et al., 2002; Kim et al., 2012)”. I would also suggest an experiment in which they measure bAP amplitude before and after adding 4AP in the bath to test whether the bAP amplitude is sensitive to block of A-type K channels. Their data in Figure 5 suggest that it would be but Krueppel suggest not. Otherwise, if Na channels have uniform density (their Figure 5), what is the mechanism for the declining bAP amplitude with distance.

We thank the reviewer for these suggestions. We have analyzed the dendritic distance in biocytin-filled GCs (278 ± 7.4 μm; n=11) and plotted it versus the normalized amplitude of the backpropagating AP. This plot is now in Figure 3E of the revised manuscript and discussed in the Results subsection “Backpropagation of axosomatic APs in the dendrites of GCs”.

The role of A-type potassium channels on bAP and dendritic spikes is now extensively discussed (Figure 5—figure supplement 1 and Figure 6—figure supplement 2) in the revised manuscript and also addressed in our answer #1 and #9.

13) The data from outside-out patches and shown in Figure 5 are very nice (and impressive) given the small size of the dendrites. However, the figure legend states that all the recordings were performed with 5 mM 4AP and 20 mM TEA in the bath. If this is correct, what is the outward current shown in 5A?

We thank the reviewer for his/her awareness on the extreme difficulty of the experiments. The small outward potassium current (~28 pA) is the resistant component to the 5 mM application of 4-AP (EC_50_= ~1 mM, see Hoffman et al., 1997, Figure 2D). We had previously seen similar outward currents in the dendrites of CA3 neurons (Kim et al., 2012, Figure 4). However, the small sodium currents present in the GC dendrites makes the outward currents more pronounced. We now mention the presence of the remaining 4-AP resistant K^+^ component in the Figure 5 legend.

14) While the authors suggest a "unique" distribution of Na and K channels (Abstract), one could argue that their results for GCs are qualitatively similar to those from CA1 pyramidal cells, but different from the conclusions of Krueppel et al. The results presented are nonetheless important since so little is known about DGC dendrites.

We agree with the reviewer and removed the word ‘unique’ from the manuscript.

[Editors' note: the author responses to the re-review follow.]

Reviewer #1:In this revision, Kim and co-authors have addressed most of my major concerns and provided new experimental data in support of the general conclusion that distal dendritic sodium-mediated spikes contribute to LTP at the perforant path (PP) synapses. The manuscript has improved and in my opinion builds a strong case for dendritic spike generation. Although the experiments are in general compelling it still contains errors and statements sometimes lack quantification (see below).Regarding the request for more detail how the authors calculated outside-out surface area the authors write, "we used Hu and Jonas, 2014". But when reviewing Figure 5 and subsection “Ionic mechanisms of AP backpropagation”, there is an obvious mistake, which was present already in the first version of this manuscript. The y-axis shows negative values for sodium conductance density, which is biophysically implausible. I have the impression that the authors rather plotted current density (unit amperes per square micrometre) but labelled their axes with conductance density (unit siemens). This error seems to propagate through the data presentation for the A-type and delayer rectifier K^+^ density distributions. I don't think that a correction for driving force was applied based on the numbers and it is neither clear whether and how area was corrected for. The authors may want to consult the article of Schmidt-Hieber and Bischofberger, 2010 (J. Neurosci. 30(30); p. 10233), read also the supplement and thoroughly revisit this part of their study. As requested earlier, it is important they spell out how data were obtained and compare their numbers with previously published data. Both for the sodium and potassium measurements.

We apologize for this mistake. We have taken the suggestion of the reviewer very seriously, and thoroughly revised this figure. To measure the membrane area of the outside-out patches and re-analyze the conductance density, we closely followed Schmidt-Hieber and Bischofberger 2010 as the reviewer suggested. First, we estimated the membrane patch area with pipettes of resistance and geometry identical to that used for our experiments in Figure 5(now presented in the new Figure 4—figure supplement 1 in the revised manuscript). We determined the linear relationship between patch area and pipette conductance under our experimental condition, resulting in A(gP) = 4.7124 x gP+ 0.21421, where A is membrane patch area (µm^2^) and gPis pipette conductance (µS). Re-analysis of our voltage-clamp recording data using our parameters revealed that overall values of conductance density are larger than those calculated by the parameters established by Hu and Jonas (2014). Nevertheless, our new results corroborate our previous conclusion that a markedly high density of K^+^ channels and a moderate and uniform density of Na^+^ channels are present in the dendrites of GCs. These data have been incorporated and are discussed in the revised manuscript in Figure 4D–F. We also have revised the Materials and methods section accordingly (subsection “Data analysis”).

Reviewer #2:Overall the authors have done a good job of addressing the points raised by the reviewers. Nevertheless, I have suggestions for additional changes to the manuscript that do not require new experiments:1) I was a little surprised to see no clear correlation between the number of putative dendritic spikes observed at the soma during TBS stimulation and the magnitude of LTP (Author response image 3). Also, the number of putative dendritic spikes associated with LTP induction seems very low. As the authors indicate this may be because they have missed detecting dendritic spikes in their somatic recordings, as suggested by the data shown in Author response image 4. The capacity to detect dendritic spikes in their somatic recordings is key to the idea that LTP induction requires dendritic spike generation, as discussed in the subsection “TBS-induced LTP at the PP-GC synapses does not require postsynaptic bAPs” and concluded from the data in Figure 1. As indicated in my original review, a characterisation of the somatic events detected when dendritic spikes are observed directly during dendritic recordings would provide direct evidence that the fast rising events observed at the soma during TBS stimulation are indeed dendritic spikes. It would also be important in my opinion to quantify how reliably dendritic spikes can be detected by somatic recordings.

We thank the reviewer for the suggestion. As the reviewer suggested, we have characterized the fast depolarizing somatic events when dendritic spikes were present (now presented in the new Figure 5—figure supplement 3 in the revised manuscript). As expected, dendritic spikes lead to a significant acceleration of the rising phase of the somatic EPSP. Interestingly, we observed that there were two groups of the fast rising somatic events, strongly and weakly propagated dendritic spikes (which is presumably due to large impedance drops at branch point). Based on these data, we again analyzed 5,200 (13 cells) somatic voltage responses to TBS induction in Figure 1. As it turns out, we identified weak (dV/dt < 10 mV/ms) and strong (dV/dt > 10 mV/ms) putative dendritic spikes that were distinguishable from EPSPs without any local regenerative events (without dendritic spikes, black, 1.3 ± 0.01 mV/ms; weak dendritic spikes, 4.1 ± 0.4 mV/ms, n = 18; Strong dendritic spikes, 14.5 ± 0.8 mV/ms; Kruskal-Wallis test: P < 0.0001; now presented in Figure 1D). Indeed, we found the strong correlation between the number of putative dendritic spikes observed during TBS and the magnitude of LTP (r = 0.77; P < 0.005; n = 13; now presented in Figure 1F). Nevertheless, we also noticed that the number of putative dendritic spikes associated with LTP induction seems low in some cells. We think that it is likely that the induction of LTP might be occurred by a single local dendritic spikes (Remy and Spruston, 2007, PNAS 104(43); p. 17192-17197).

2) Some of the supplemental data is key to the story in my opinion. In particular, I am referring to the data in Figure 1—figure supplements 1 and 2. I suggest parts (or all) of Figure 1—figure supplement 1 is included in Figure 1, and parts (or all) of Figure 1—figure supplement 2 is included in Figure 2.

We have moved previous Figure 1—figure supplements 1 and 2 into the main body (Figure 1D and 2B, respectively) of the manuscript as suggested.

3) I still think the rational for Figure 4 is weak. This figure does not add much to the paper. The observed bAP conduction velocities are not that different from that seen in pyramidal cells where standard +10/-10 ms EPSP-AP timing protocols can evoked STDP at proximal synapses.

We agree with the reviewer that Figure 4 in the original manuscript could be confusing to readers that the conduction velocity of bAPs would cause the different plasticity form in GCs. Following the suggestion of the reviewer, we decided to remove this Figure 4. We also have revised the text accordingly (subsection “LTP by TBS at PP–GC synapses requires NMDARs and Na^+^ channels”).

[Editors’ note: the author responses to the re-review follow.]

Specific points:1) There is a little confusion in the first section of the Results ("TBS-induced LTP at the PP-GC synapses does not require postsynaptic bAPS"). The authors first state "To ensure that no axosomatic AP initiation and backpropagation occur during TBS, we locally applied tetrodotoxin (TTX) to the GC axon, soma, and proximal dendrites in a subset of experiments (6 out of 13 experiments)", then later you say "To test the contribution of bAPs in this form of LTP, we applied strong synaptic stimulation without perisomatic TTX application". It would be better to first discuss and show the control case in the absence of TTX, describe as you do the presence of putative dendritic spikes during synaptic stimulation and how these are correlated with the magnitude of LTP, and only then introduce the idea that bAPs are not required showing both that the magnitude of LTP is not related to the presence of APs under control conditions and that LTP persists when APs were blocked.

Following the suggestion of the reviewer closely, we presented the data of TBS-induced LTP experiments in two separate sets upon whether axosomatic APs are present or absent during TBS (Figure 1B, with axonal firing, n = 8; Figure 1C, without axonal firing, n = 13). In Figure 1C, we either used perisomatic TTX application or adjusted the stimulus intensity so that high-frequency PP stimulation would not trigger axosomatic APs during TBS.

Accordingly, we have moved the data in previous Figure 1—figure supplement 1 (showing that APs are not required for LTP) into the main Figure (now in Figure 1B in the revised manuscript), and have substantially revised Figure 1 and the corresponding the section. We hope that the reviewer will find the revised section improved.

2) The finding that APV "abolished LTP”, but had little if any impact on putative dendritic spikes (Figure 2B and subsection “LTP by TBS at PP–GC synapses requires NMDARs and Na^+^ channel”: "fast rising events remained unchanged") questions the causal role of these putative dendritic spikes in LTP induction. The authors address this by using QX-314 to block Na^+^ channels, finding that this "abolished both plateau potentials and TBS-induced LTP". Do the authors mean "abolished both putative dendritic spikes and TBS-induced LTP"? Assuming this is the case some quantification of the effect of QX-314 on putative dendritic spikes is warranted (E.g. data on the number of putative dendritic spikes in control versus QX-314). Note, also the figure reference in the aforementioned subsection should be Figure 2C, D not Figure 2B-D.

We have corrected the sentence as suggested. We have added the new bar plots comparing the baseline EPSP amplitudes in two experimental groups (EPSP_control_: 7.05 ± 0.54 mV, n = 7; EPSP_QX-314_: 9.24 ± 1.21 mV, n = 7; *P* = 0.10). We also have provided the data on the number of putative dendritic spikes in control versus QX-314, indicating that intracellular application of QX-314 abolished dendritic spike initiation during TBS (control: 4.14 ± 1.06, n = 7; QX-314: 0, n = 7; *P* < 0.005). It is now documented in Figure 3D and discussed the revised manuscript.

3) The authors state that "Pooled data demonstrated that a moderate density of Na^+^ channels is distributed over the dendritic membrane", yet then indicate dendritic Na^+^ current densities of between 136 pS/um^2 (proximal) and 206 pS/um^2 (distal). These are high not moderate densities. One model of AP backpropagation in granule cells Krueppel et al. (2011) used dendritic Na^+^ channel densities of around 2 mS/cm^2 (or 20 pS/um^2). That is, almost a factor of 10 lower than estimated by the authors. The estimated densities of A-type and delayed rectifier type K^+^ channels are also very high. Given some uncertainty in the accuracy of estimating the patch membrane area based on pipette capacitance, it might therefore be better to simply state the peak current amplitude rather than current density.

We acknowledge the suggestion of the reviewer and have changed all the conductance density values to the peak current amplitudes, which now presented in the new Figure 5D, E, and F in the revised manuscript. As we used similar glass micropipettes (i.e., resistance and geometry) in both somatic and dendritic recordings, these data will provide reliable comparisons of ion channel densities in the soma and the dendrites. We also have revised the Materials and methods section accordingly.

4) Addition of the data showing the association between putative dendritic spikes observed at the soma (based on dV/dt) and the direct observation of dendritic spikes during dendritic recordings is most welcome. In my view this data (Figure 5—figure supplement 3) is critical to the paper and therefore should not be "buried" in the supplemental data, but should be included as part of Figure 5 of the manuscript. I would argue the data in Figure 1—figure supplement 1, showing that APs are not required for LTP, is also critical to the story and therefore should also be presented as a main figure rather than supplemental data.

We appreciate the suggestion of the reviewer. We agree with the reviewer that the association between putative dendritic spikes observed at the soma and the direct observation of dendritic spikes during dendritic recordings is important information of the manuscript and therefore should be presented in the main body of the manuscript. Accordingly, we have moved previous Figure 5—figure supplement 3 into Figure 6 in the revised manuscript. We also have revised the text accordingly. Please see our answer #1 about the data in previous Figure 1—figure supplement 1.

[Editors' note: further revisions were requested prior to acceptance, as described below.]

The manuscript has been improved but there are a few remaining issues that need to be addressed before acceptance, as outlined below:1) There is some confusion regarding the figure labeling in the Results text, especially around new Figure 5 (formerly Figure 4, subsection “Ionic mechanisms of AP backpropagation”). Please check carefully to ensure that figures are cited correctly.

We apologize for this error. We have corrected this problem as suggested (subsection “Ionic mechanisms of AP backpropagation”).

2) Although strong dendritic spikes are defined as > 10 mV/ms in the subsection “TBS-induced LTP at the PP-GC synapses does not require postsynaptic bAPs”, the definition of weak dendritic spikes is more ambiguous, especially given that some thresholds must have been set to distinguish between EPSPs and weak dendritic spikes. This discriminant belongs in the text. The text table in Figure 7B suggests it is on the order of 3 mV/ms as the minimum somatic detection of a DS.

We thank the reviewer for raising this important point. EPSPs with or without dendritic spikes could be readily distinguished by the spikelet waveforms. However, we agree with the reviewer that our definition of weak putative dendritic spikes is not clear. To avoid ambiguity, it would be much helpful to set the threshold level of d*V*/d*t* to be counted as dendritic spikes. Therefore, we have once again carefully analyzed the d*V*/d*t* of somatic voltage responses, and found that d*V*/d*t* of the EPSPs that show a distinct spikelet waveform is larger than 2.5 mV/ms. We have added this value as the criteria for somatic detection of weak putative dendritic spikes as requested (subsection “TBS-induced LTP at the PP-GC synapses does not require postsynaptic bAPs”, first paragraph).

Once again, we would like to thank the reviewers for their time and their careful analysis of our manuscript, which helped us to further improve our paper.